# Explainable Voting

**Dominik Peters**
Harvard University
dpeters@seas.harvard.edu

**Ariel D. Procaccia**
Harvard University
arielpro@seas.harvard.edu

**Alexandros Psomas**
Purdue University
apsomas@cs.purdue.edu

**Zixin Zhou**
CFCS, Peking University
zhouzixin1998@gmail.com

## Abstract

The design of voting rules is traditionally guided by desirable axioms. Recent work shows that, surprisingly, the axiomatic approach can also support the generation of explanations for voting outcomes. However, no bounds on the size of these explanations is given; for all we know, they may be unbearably tedious. We prove, however, that outcomes of the important Borda rule can be explained using $O(m^2)$ steps, where $m$ is the number of alternatives. Our main technical result is a general lower bound that, in particular, implies that the foregoing bound is asymptotically tight. We discuss the significance of our results for AI and machine learning, including their potential to bolster an emerging paradigm of automated decision making called virtual democracy.

## 1 Introduction

Voting plays a foundational role in many settings and guises, such as when electing leaders, when aggregating expert recommendations, or when merging the outputs of statistical models. Unfortunately, the voting method that is the most popular is also the worst: *plurality voting*, which simply selects the alternative that is the favorite of the highest number of voters (or experts, models). This method is bad because it routinely selects low-quality options [Merrill, 1984, Bordley, 1983] and because it distorts incentives [e.g., Tideman, 1987]. This makes it the least popular rule in surveys among voting theorists [Laslier, 2012].

Researchers and practitioners have developed alternative voting schemes that produce higher-quality decisions, by eliciting not just each voter's top choice, but a ranking of all options. While theoretically superior, these methods have seen limited uptake in user-facing systems. One reason is that these rules can be opaque and abstract. Rules are typically "explained" to users through pseudocode, and often, the only explanation of the voting outcome that is provided to users (if any!) is a trace of the algorithm's execution. Sure, this allows voters to verify that the algorithm was applied correctly, but it is unlikely to fill them with confidence in the quality of the public decision. Our aim is to develop methods that can provide an explanation of the decision that is not merely procedural, but based on intuitively compelling principles that characterize good decision making.

**Applications**   Voting is often a component of machine learning systems, such as in ensemble learning (where a combination rule is used to merge the labels of different models) or in the emerging paradigm of *virtual democracy*, an approach to automated decision making. The idea of the latter approach is to collect preference data from a group of voters, and use it to learn models of their preferences over a (possibly infinite) set of alternatives. At runtime, when a specific set of alternatives is presented, the system makes a decision by applying a voting rule to the *predicted* preferences

of the voters over the current alternatives. This approach has led to proof-of-concept systems that automate moral decisions faced by autonomous vehicles [Noothigattu et al., 2018] and kidney exchanges [Freedman et al., 2018]. Most notably, virtual democracy is the foundation of a pilot recommendation system used to allocate food donations to recipient organizations [Lee et al., 2019].

The voting rule implemented in most virtual democracy applications is the *Borda rule*, in part because it was shown to be robust to the type of prediction errors that arise when using preference models [Kahng et al., 2019]. Under the Borda rule — which dates back to the 18th Century — each voter gives $m - k$ points to the alternative she ranks in the $k^{\text{th}}$ place, where $m$ is the number of alternatives; winning alternatives maximize the overall score. Since it is widely used, we focus on this rule, but our framework applies to other rules (notably, approval voting) as well.

**Our Approach**   For classifiers, techniques for computing explanations have advanced rapidly, making classification more transparent. However, those techniques are not enough for systems that combine outputs from many models, such as virtual democracy. To explain these composed systems, we not only need to explain the individual models, but must also explain how we merged their outputs to obtain a final decision.

An immediate difficulty for our task is that most of the principles we might like to use in our explanations are too weak. For example, the *unanimity axiom* says that if each voter ranks $a$ in top position, then $a$ should win. This is uncontroversial, but in practice we almost never see such profiles, making it useless for explanations. However, there is another class of principles that are more powerful: *consistency axioms*, which constrain the voting rule to make compatible decisions in different situations. For example, the *reinforcement axiom* requires that if $a$ wins when aggregating the first half of the votes, and $a$ also wins when aggregating the other half of the votes, then $a$ should win when aggregating all votes simultaneously.

While the reinforcement axiom alone is not powerful enough to explain Borda outcomes, Cailloux and Endriss [2016] suggest that a *chain* of axioms would do the trick. Given a preference profile, *intraprofile axioms* that pertain to a single profile (like unanimity) can be applied to convince a human of the outcome on certain profiles; and *interprofile axioms* that connect several profiles (like reinforcement) can be used to relate the outcome on these profiles to the outcome on other profiles, in a way that ultimately yields the desired outcome on the profile at hand.

For example, consider a trivial preference profile that only includes one voter with the ranking $(a, b, c)$. Since $a$ has unanimous support, the unanimity axiom designates it as the winner. Another profile has two voters associated with the rankings $(a, b, c)$ and $(c, b, a)$ (notice that the latter ranking is the reverse of the former); by an intraprofile axiom called *cancellation*, all alternatives must be tied under this profile. Now, due to reinforcement, the winner on the combined profile (that contains all three voters) must be the intersection of the two sets of winners, namely $a$. This is, in fact, the alternative chosen by the Borda rule. Therefore, we can explain the outcome of Borda on the combined profile by chaining three axioms together. Notice that this reasoning is essentially given in natural language, and it is clearly possible to automatically transform it into an explanation that a layperson can understand.

Cailloux and Endriss [2016] prove that for any profile, the outcome under Borda can be explained through a chain of axioms involving six different axioms. This is an inspiring result — but it is unclear whether it is practical, as the *length* of the explanation is unknown. An explanation with a thousand steps is not something a user will sit through. Our research challenge, therefore, is to generate the shortest possible explanations of voting outcomes — and analyze the required length — while extending the approach of Cailloux and Endriss [2016] beyond the Borda rule.

The positive result of Cailloux and Endriss [2016] is based on an earlier result by Young [1974], who proved that Borda is the unique voting rule satisfying the six axioms in the generated explanations. Given that theorem, one might be tempted to justify the voting procedure by a simple reference to the Young paper. However, our approach is more powerful: due to Young [1974], we can promise users in advance what properties the voting outcome will satisfy; then, once we have determined the winner, we can show users that our promise was fulfilled, by presenting an explanation. Without an explanation, users would either need to be experts in linear algebra, or else take our word that the voting rule is determined by the axioms.

We expect explainable voting to help with the broader adoption of automated decision making systems. Whether virtual democracy systems directly make decisions or merely support decisions by making

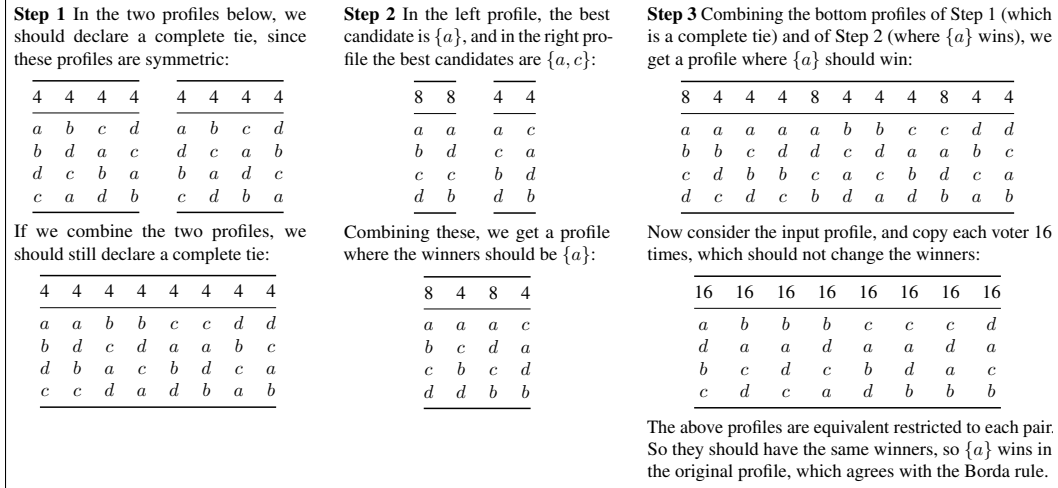

Figure 1: An explanation of the Borda outcome on an example profile. The column header shows how many voters submit each ranking.

recommendations, they must be trusted by stakeholders in order to be adopted. Lee et al. [2019] provide evidence that the participatory nature of voting-based systems inherently encourages trust. Still, the capability to provide meaningful explanations of decisions or recommendations obtained via voting would amplify that trust. Taking a broader perspective, explainable voting is applicable to the myriad environments where voting is employed, from ensemble learning through online services like RoboVote.org all the way to — dare we say it? — political elections.

**Our Results** We propose giving multi-step explanations of voting outcomes. Mathematically, it is convenient to model these explanations as *formal proofs* in an axiom system consisting of the voting principles we are interested in. Written in formal logic, these explanations look impenetrable. Written in natural language, they are accessible and easy to follow, since the axioms have been optimized for intuitive appeal. We show an example output of the system in Figure 1 to explain that Borda selects $a$ as the unique winner in the eight-voter profile $(a, d, b, c), (b, a, c, d), (b, a, d, c), (b, d, c, a), (c, a, b, d), (c, a, d, b), (c, d, a, b), (d, a, c, b)$. In Appendix A we provide an additional example based on election data from the 2009 mayoral election in Burlington, Vermont, with more than 3,000 voters; notably, the resulting explanation is barely longer than the one shown in Figure 1.

In Section 3 we focus on the Borda rule, which — as we mentioned above — has special significance. Building on the work of Cailloux and Endriss [2016], we introduce seven natural axioms that characterize Borda. We then prove (Theorem 1) that, in this framework, Borda outcomes can always be explained in $O(m^2)$ steps, where $m$ is the number of alternatives. This result gives an algorithm for automatically generating explanations like in Figure 1. It is practical in settings with a small to moderate number of alternatives.

In Section 4 we lay the groundwork for, and prove, our main result: a general lower bound on the length of explanations. The key idea behind our approach is to *embed* voting rules into linear spaces, which allows us to apply linear algebra machinery. For example, Borda outcomes can be determined purely from the fraction of voters who prefer one alternative to another for every pair of alternatives, and therefore Borda can be embedded into $\mathbb{Q}^{\binom{m}{2}}$. The lower bound, Theorem 2, depends on the dimension of the linear space A notable aspect of this result is that it holds not just in the worst case, but for almost every profile. As corollaries we get asymptotically tight lower bounds for Borda as well as two other rules, plurality and approval. Our lower bounds guide the way towards voting rules whose outcomes can be easily explained even when there are many alternatives — a point that we discuss in Section 5.

## 2 Preliminaries

In this section we provide some social choice terminology, and describe the framework of Cailloux and Endriss [2016] for the explanation of voting outcomes.

### 2.1 Basic Terminology

Let $\mathcal{A}$ be a finite set of alternatives and denote $m = |\mathcal{A}|$. Let $\mathcal{P}_\emptyset(\mathcal{A})$ be the set of non-empty subsets of $\mathcal{A}$. Preferences of voters are given by (strict) rankings over $\mathcal{A}$; let $\mathcal{A}!$ be the set of strict rankings. A *preference profile* (or simply *profile*) is a function $\mathbf{R} : \mathcal{A}! \to \mathbb{N}$ that specifies how many voters report each possible ranking.[1] Let $\mathcal{R}$ be the set of all non-empty profiles, that is, all profiles except the one that maps all $m!$ rankings to zero. A *voting rule* $f : \mathcal{R} \to \mathcal{P}_\emptyset(\mathcal{A})$ maps each profile $\mathbf{R} \in \mathcal{R}$ to a non-empty subset of $\mathcal{A}$, the set of tied winners for $\mathbf{R}$. For two profiles $\mathbf{R}_1$ and $\mathbf{R}_2$, we define $\mathbf{R}_1 \bigoplus \mathbf{R}_2$ as the sum of the two profiles (that is, the multiplicity of each ranking is the sum of its multiplicities in the two profiles). For $k \in \mathbb{Z}_+$, we define $k\mathbf{R}$ as the sum of $k$ copies of $\mathbf{R}$.

We will pay special attention to the Borda rule. As mentioned in Section 1, under Borda each voter awards $m - k$ points to the alternative ranked in the $k^{\text{th}}$ position; the winner set consists of all alternatives with maximum score. For example, if the votes are $a \succ b \succ c \succ d$, $d \succ b \succ c \succ a$, and (again) $d \succ b \succ c \succ a$, then the winner set would be $\{b, d\}$, as both alternatives have 6 points.

### 2.2 Explainability Framework

We will produce explanations as proofs in a language of propositional logic over propositional variables (or atomic formulae) $\{[\mathbf{R} \mapsto A] : \mathbf{R} \in \mathcal{R}, A \in \mathcal{P}_\emptyset(\mathcal{A})\}$. The language $\mathcal{L}$ is the set of all formulae that can be formed using these variables and logical connectives $\neg, \wedge, \vee, \to$.

A voting rule $f$ induces a truth assignment $v_f$ to the propositional variables which assigns value *true* to atom $[\mathbf{R} \mapsto A]$ if $f(\mathbf{R}) = A$ and value *false* otherwise. By standard semantics of propositional connectives, this truth assignment extends to all formulae of $\mathcal{L}$. A truth assignment $v$ *satisfies* a set of formulae if $v$ assigns *true* to all formulae in the set.

We can translate familiar axioms for voting rules in social choice theory into the language $\mathcal{L}$. An *$\mathcal{L}$-axiom* is a set of formulae, each of which we call an *axiom instance*. For example, the unanimity axiom can be written as $\{[\mathbf{R} \mapsto \{a\}] : a \in A, \mathbf{R} \in \mathcal{R} \text{ and every voter in } \mathbf{R} \text{ ranks } a \text{ top}\}$.

A basic axiom is **FUNC** which requires that $f$ assigns exactly one set $A$ to each profile $\mathbf{R}$. Thus, **FUNC** consists of the formulae $\bigvee_{A \in \mathcal{P}_\emptyset(\mathcal{A})} [\mathbf{R} \mapsto A]$ and $\bigwedge_{A_1 \neq A_2} \neg [\mathbf{R} \mapsto A_1] \vee \neg [\mathbf{R} \mapsto A_2]$ for each $\mathbf{R} \in \mathcal{R}$. As a background assumption, we will usually not explicitly mention **FUNC**.

A voting rule $f$ *satisfies* an $\mathcal{L}$-axiom **X** if $v_f$ satisfies **X** (recall that an $\mathcal{L}$-axiom is a set of formulae). An *$\mathcal{L}$-axiomatization* is a set $\mathcal{S}$ of $\mathcal{L}$-axioms. With a slight abuse of terminology, a voting rule $f$ satisfies an $\mathcal{L}$-axiomatization $\mathcal{S}$ if $f$ satisfies every axiom in $\mathcal{S}$. Finally, $\mathcal{S}$ *characterizes* a voting rule $f$ if and only if $f$ is the only voting rule satisfying every axiom in $\mathcal{S}$.

Let $\mathcal{S}$ be an $\mathcal{L}$-axiomatization. An *explanation* of an outcome $A$ for profile $\mathbf{R}$ in terms of $\mathcal{S}$ is a formal proof of the formula $[\mathbf{R} \mapsto A]$ in a suitable proof system for propositional logic, using axioms in $\mathcal{S}$ as assumptions. Any sound and complete proof system will work, but for concreteness let us define a *proof of formula $\varphi$ assuming $\mathcal{S}$* to be a sequence $\varphi_1, \ldots, \varphi_r = \varphi$ of propositional formulae such that for each $i = 1, \ldots r$, we have that either (i) $\varphi_i$ is an instance of an axiom in $\mathcal{S}$, or (ii) $\varphi_i$ is an instance of an axiom in **FUNC**, or (iii) $\varphi_i$ is a tautology (a formula that is satisfied by every variable assignment), or (iv) there exist $j, k < i$ such that $\varphi_k = \varphi_j \to \varphi_i$ (*modus ponens*). This is a type of Hilbert system and is sound and complete [see, e.g., Hamilton, 1988, Section 2.2]. The *length* of the proof is the number $r$ of formulae in the sequence. For ease of exposition, when writing down proofs in this system, we will often skip steps that use only propositional reasoning.

# 3 An Upper Bound for Borda

In this section we present our upper bound on the length of explanations required by the Borda rule using a particular, natural axiomatization. Specifically, we show that an explanation of length $O(m^2)$ suffices; as we will see in Section 4, our main result implies that this is optimal.

We start by defining families of profiles that are useful in producing short proofs. The first family consists of *elementary* profiles $\mathbf{R}_{\text{elem}}^A$, for each non-empty $A \subseteq \mathcal{A}$, which have two voters. Let $A = \{x_1, \ldots, x_k\}$ and $\mathcal{A} \setminus A = \{y_1, \ldots, y_{m-k}\}$. The first voter has preferences $x_1 \succ x_2 \succ \cdots \succ x_k \succ y_1 \succ \cdots \succ y_{m-k}$. The second voter has preferences $x_k \succ \cdots \succ x_1 \succ y_{m-k} \succ \cdots \succ y_1$. For example, the elementary profile $\mathbf{R}_{\text{elem}}^{\{a,b\}}$, when $\mathcal{A} = \{a, b, c, d\}$, has two votes: $a \succ b \succ c \succ d$ and $b \succ a \succ d \succ c$. Intuitively, in the profile $\mathbf{R}_{\text{elem}}^A$, the alternatives in $A$ are similar to each other (since the preferences over $A$ 'cancel'), and stronger than the other alternatives, so a sensible voting rule should select the alternatives in $A$. The second family consists of *cyclic* profiles $\mathbf{R}_{\text{cyc}}^T$, where $T$ is an $m$-cycle over alternatives, which is composed of all rankings generated by $T$. For example, the cyclic profile $\mathbf{R}_{\text{cyc}}^{\langle a,b,c,a\rangle}$ contains the rankings $(a, b, c)$, $(b, c, a)$ and $(c, a, b)$. Intuitively, by symmetry, a sensible voting rule should declare a tie between all alternatives in a cyclic profile.

We also require the notion of the *delta* vector $\delta^{\mathbf{R}}$ of a profile $\mathbf{R}$, which is a vector with $\binom{m}{2}$ coordinates, where $\delta_{ab}^{\mathbf{R}}$ is the number of voters who prefer alternative $a$ to alternative $b$ minus the number of voters who prefer $b$ to $a$. The delta vector consists of the majority margins of the profile $\mathbf{R}$. For example, if $\delta_{ab}^{\mathbf{R}} > 0$, then a majority of voters prefers $a$ to $b$.

An important observation is that the delta vector is a sufficient statistic for computing the outcome under Borda. Indeed, for an alternative $a \in \mathcal{A}$, one can check that the Borda score of $a$ is equal to $\frac{1}{2}\sum_{b\in\mathcal{A}} \delta_{ab}^{\mathbf{R}} + n(m-1)/2$, where $n$ is the total number of voters in $\mathbf{R}$. In any fixed $\mathbf{R}$, the second term is constant, and so we can find Borda scores and winners by inspecting only the delta vector.

We are now ready to define the axioms we need for our upper bound. We use the axiomatization proposed by Cailloux and Endriss [2016]. The first three axioms give single-step proofs for "base profiles." These are the *intraprofile* axioms. (Formally, each of these axioms is a set of formulas.)

1. **ELEM**: For each elementary profile $\mathbf{R}_{\text{elem}}^A$, the set of winners should be $A$, so $\left[\mathbf{R}_{\text{elem}}^A \mapsto A\right]$.

2. **CYCL**: For each cyclic profile $\mathbf{R}_{\text{cyc}}^T$, the set of winners should be all of $\mathcal{A}$, so $\left[\mathbf{R}_{\text{cyc}}^T \mapsto \mathcal{A}\right]$.

3. **CANC**: If for all $a, b \in \mathcal{A}$, the same number of voters prefer $a$ to $b$ as prefer $b$ to $a$, then the set of winners is $\mathcal{A}$. Formally, for all $\mathbf{R}$ with $\delta_{ab}^{\mathbf{R}} = 0$ for all $a, b \in \mathcal{A}$, $[\mathbf{R} \mapsto \mathcal{A}]$.

The remaining axioms are *interprofile* axioms, linking outcomes between different profiles. The first axiom captures reinforcement. The others capture consequences of reinforcement; making them separate axioms gives us convenient shortcuts in the generated explanations.

4. **REINF**: For any two profiles $\mathbf{R}_1$ and $\mathbf{R}_2$, and any two subsets of alternatives $A_1$ and $A_2$ with $A_1 \cap A_2 \neq \emptyset$, it holds that $([\mathbf{R}_1 \mapsto A_1] \wedge [\mathbf{R}_2 \mapsto A_2]) \to [\mathbf{R}_1 \bigoplus \mathbf{R}_2 \mapsto A_1 \cap A_2]$.

5. **REINF-SUB**: Subtracting a profile with a full winner set does not change the outcome. Formally, for all $\mathbf{R}, \mathbf{R}'$, $([\mathbf{R} \bigoplus \mathbf{R}' \mapsto A] \wedge [\mathbf{R}' \mapsto \mathcal{A}]) \to [\mathbf{R} \mapsto A]$.

6. **SIMP**: A profile that is a repetition of some sub-profile should have the same set of winners as the sub-profile. Formally, for each $k \in \mathbb{Z}_+$, $[k\mathbf{R} \mapsto A] \to [\mathbf{R} \mapsto A]$.

7. **MULT**: If a profile $\mathbf{R}$ has winner set $A$, then the profile that repeats $\mathbf{R}$ $k$ times has the same winner set. Formally, for each $k \in \mathbb{Z}_+$, $[\mathbf{R} \mapsto A] \to [k\mathbf{R} \mapsto A]$.

The conjunction of **SIMP** and **MULT** is known as *homogeneity*. Cailloux and Endriss [2016] did not use **MULT**; we add it for convenience. Let us refer to the $\mathcal{L}$-axiomatization consisting of Axioms 1–7 listed above as $\mathcal{S}_{\text{Borda}}$. Cailloux and Endriss [2016] show that $\mathcal{S}_{\text{Borda}}$ characterizes the Borda rule (based on a result of Young [1974]) and that for any profile $\mathbf{R}$, the outcome of Borda can be explained using $\mathcal{S}_{\text{Borda}}$ (and no other outcome can be so explained). Technically, this means that given a profile $\mathbf{R}$, it is possible to give a proof that the atomic formula $\varphi = [\mathbf{R} \mapsto f(\mathbf{R})]$ is such that $v_f(\varphi) = true$ for all voting rules $f$ satisfying $\mathcal{S}_{\text{Borda}}$ (and Borda is the only such rule). Our first theorem strengthens this existence result by bounding the length of the required explanation. We only give a rough proof sketch here to outline the strategy, and leave the details to Appendix C.1.

**Theorem 1.** *For any profile* $\mathbf{R}$ *with* $m$ *alternatives, the outcome of the Borda rule can be explained in* $O(m^2)$ *steps assuming the* $\mathcal{L}$*-axiomatization* $\mathcal{S}_{\text{Borda}}$.

*Proof sketch.* The linear space $\mathbb{Q}^{\binom{m}{2}}$ of delta vectors is spanned by the delta vectors induced by elementary and cyclic profiles. Given a profile $\mathbf{R}$, we can find another profile $\mathbf{R}'$ which is a linear combination of at most $O(m^2)$ different elementary and cyclic profiles, satisfying $k\delta^{\mathbf{R}} = \delta^{\mathbf{R}'}$ for some $k \in \mathbb{Z}_+$. By the latter equality, $\mathbf{R}$ and $\mathbf{R}'$ have the same set of Borda winners. Using **ELEM**, **CYCL**, and interprofile axioms, we can show that $f$ must elect the Borda winners at $\mathbf{R}'$. Using **CANC** and interprofile axioms, we can show that since $k\delta^{\mathbf{R}} = \delta^{\mathbf{R}'}$, we must have $f(\mathbf{R}) = f(\mathbf{R}')$, which together gives an explanation of the Borda outcome at $\mathbf{R}$. The length of the explanation is determined by the length of the decomposition of $\mathbf{R}'$, which is in $O(m^2)$. $\qquad\square$

# 4 A General Lower Bound

In this section we prove our main result: a general lower bound on the required explanation length, which applies to a broad class of axiomatizations. Detailed proofs appear in Appendix B, where, in fact, we prove a stronger version which enriches the model with axioms based on *linear predicates*. This extra power is not needed for the Borda rule but is helpful to capture other voting rules.

## 4.1 Mathematical Framework

The Borda rule depends only on the delta vector, and the explanations constructed in Theorem 1 exploit the linear algebra of delta vectors. Our result applies more generally to voting rules that can be embedded into a linear space, and axiomatizations based on the embedding. All definitions in this section are novel, to the best of our knowledge. Some structures that appear in the study of compilation complexity of voting rules [Chevaleyre et al., 2009] have a similar flavor.

**Definition 1.** *A voting rule* $f : \mathcal{R} \to \mathcal{P}_\emptyset(\mathcal{A})$ *can be embedded into a linear space* $V$ *over* $\mathbb{Q}$ *via* $h : \mathcal{R} \to V$ *and* $g : V \to \mathcal{P}_\emptyset(\mathcal{A})$ *if the following properties are satisfied:*

1. $h(\mathbf{R} \bigoplus \mathbf{R}') = h(\mathbf{R}) + h(\mathbf{R}')$ *for all* $\mathbf{R}, \mathbf{R}' \in \mathcal{R}$.[2]

2. $f(\mathbf{R}) = g(h(\mathbf{R}))$ *for all* $\mathbf{R} \in \mathcal{R}$.

3. $\{h(\mathbf{R}) : \mathbf{R} \in \mathcal{R}\}$ *spans* $V$.

For example, any (anonymous) voting rule $f$ can be trivially embedded into a linear space of dimension $m!$: the vector $h(\mathbf{R})$ shows how often each preference ranking appears in $\mathbf{R}$, and $g$ maps each possible such vector to a set of winners. The Borda rule can be embedded into $\mathbb{Q}^{\binom{m}{2}}$ by $h(\mathbf{R}) = \delta^{\mathbf{R}}$ and $g(\delta) = \arg\max_{a \in \mathcal{A}} \sum_{b \in \mathcal{A}} \delta_{ab}$. One could also embed the Borda rule into $\mathbb{Q}^m$, with $h$ returning the vector of Borda scores, and $g$ selecting the alternatives with highest score.

**Definition 2.** *Let* $f$ *be a voting rule that is embedded into a linear space* $V$ *by* $h$ *and* $g$. *Let* $\circ : \mathcal{P}_\emptyset(\mathcal{A}) \times \mathcal{P}_\emptyset(\mathcal{A}) \to \mathcal{P}_\emptyset(\mathcal{A})$ *be a binary operation. We say that* $g$ *admits the operation* $\circ$ *if* $g(v + v') = g(v) \circ g(v')$ *for all* $v, v' \in V$ *such that* $g(v) \circ g(v') \neq \emptyset$.

Intuitively, the operation $\circ$ describes how to combine two voting outcomes. For the examples above, if the rule $f$ satisfies reinforcement, then the embedding admits the operation $\circ = \cap$.

We now describe an $\mathcal{L}$-axiomatization for an embedded voting rule. This axiomatization has several abstract components, which we will discuss further after the definition.

**Definition 3.** *Let* $f$ *be a voting rule that can be embedded into a linear space* $V$ *by* $h$ *and* $g$, *and assume that* $g$ *admits operation* $\circ$, *which is commutative. Let* $S \subseteq \mathcal{R}$ *be a set of* base profiles, *such that* $S$ *can be written as a finite union of sets of profiles,* $S = \bigcup_{k=1}^{N} S_i$, *for some* $N$, *where each* $S_i \subseteq \mathcal{R}$ *is a possibly infinite set of profiles, and* $h(S_i)$ *lies in a* one-dimensional *subspace of* $V$. *Then the* $\mathcal{L}$*-axiomatization* $\mathcal{S}(f, h, g, \circ, V, S)$ *consists of the following three axioms:*

1. **ADD**: *For all* $\mathbf{R}_1, \mathbf{R}_2 \in \mathcal{R}$ *such that* $A_1 \circ A_2 \neq \emptyset$,
   $[\mathbf{R}_1 \mapsto A_1] \wedge [\mathbf{R}_2 \mapsto A_2] \to [\mathbf{R}_1 \bigoplus \mathbf{R}_2 \mapsto A_1 \circ A_2]$.

2. **EMB**: *For all* $\mathbf{R}_1, \mathbf{R}_2 \in \mathcal{R}$ *such that* $h(\mathbf{R}_1) = h(\mathbf{R}_2)$,
   $[\mathbf{R}_1 \mapsto A] \to [\mathbf{R}_2 \mapsto A]$.

3. **INIT**: *For all* $\mathbf{R} \in S$,
   $[\mathbf{R} \mapsto f(\mathbf{R})]$.

If $\circ = \cap$ then **ADD** is simply reinforcement. The **INIT** axioms are intraprofile axioms that prescribe the outcome on the set $S$ of base profiles; this is similar to the intraprofile axioms we saw for Borda, where $S$ would consist of elementary, cyclic, and cancellation profiles. **EMB** encodes the fact that if a voting rule $f$ is embedded into $V$ by an embedding $h, g$, then $f$ must have the same outcome on $\mathbf{R}_1$ and $\mathbf{R}_2$ if $h(\mathbf{R}_1) = h(\mathbf{R}_2)$. For example, when embedding Borda using delta vectors, this axiom would say that two profiles with the same delta vector must yield the same outcome.

Axiomatizations like the one we gave for Borda in Section 3 do not follow the format of Definition 3 precisely. To apply our results, the following relaxation is helpful. It generalizes the axiomatizations of Definition 3, and our lower bounds still apply.

**Definition 4.** *An axiomatization $\mathcal{S}$ of a voting rule $f$ is* asymptotically weaker *than $\mathcal{S}_{\mathrm{emb}} = \mathcal{S}(f, h, g, \circ, V, S)$ if there is $c \geq 1$ such that for every axiom instance $\varphi$ in $\mathcal{S}$, there exists a proof of $\varphi$ assuming $\mathcal{S}_{\mathrm{emb}}$ that uses at most $c$ axiom instances of form **INIT**, as well as an unlimited number of **ADD** and **EMB** axiom instances.*

For example, for any $k \in \mathbb{Z}_+$, $\mathcal{S}$ could include the axiom

$$[\mathbf{R} \mapsto A] \to \left[ k\mathbf{R} \mapsto \underbrace{A \circ A \circ \ldots \circ A}_{k \text{ times}} \right]$$

since it can be deduced by repeatedly using **ADD**.

## 4.2 Theorem Statement and Proof

In this section we give a lower bound on the length of explanation required for random profiles. Specifically, assume that the preferences of voters are independent, and each voter picks their ranking over all $m!$ possibilities uniformly at random. This common assumption is known as the *impartial culture assumption* in social choice theory [Tsetlin et al., 2003]. Let $\mathbf{R}^n$ denote the random profile generated this way for the case of $n$ voters. When we say that $\mathbf{R}^n$ satisfies a property "with high probability," we mean that the probability converges to 1 as $n$ goes to infinity.

**Theorem 2.** *Let $f$ be a voting rule that can be embedded into a linear space $V$ of finite dimension $d$ by $h$ and $g$. Consider an axiomatization $\mathcal{S}$ of $f$ that is asymptotically weaker than some axiomatization $\mathcal{S}(f, h, g, \circ, V, S)$ based on operation $\circ$ and base profiles $S$ satisfying the conditions of Definition 3. Then, with high probability, every explanation of the outcome $f(\mathbf{R}^n)$ at the random profile $\mathbf{R}^n$ using $\mathcal{S}$ consists of $\Omega(d)$ steps.*

The impartial culture assumption is debatable as a model of voter preferences, but in our case that objection is not relevant: because the theorem's conclusion holds with high probability, it provides a lower bound with respect to *almost every profile*. Moreover, the proof can be adapted to work for any $\mathcal{D}$ over $\mathcal{R}$ such that $h(\mathrm{supp}(\mathcal{D}))$ spans $V$; we focus on impartial culture for ease of exposition.

The high-level idea of the theorem's proof is as follows. We are going to show that for large enough $n$, if we only use a set $\mathcal{B}$, $|\mathcal{B}| = d - 1$, of axioms from **INIT** (and as many axioms as we want from **ADD** and **EMB**) in the explanation of $[\mathbf{R}^n \mapsto f(\mathbf{R}^n)]$, then with high probability, for every such $\mathcal{B}$ there is another voting rule $f'$, which satisfies all axioms in **ADD**, **EMB**, and in $\mathcal{B}$, such that $f'$ disagrees with $f$ on $\mathbf{R}^n$, i.e. $f'(\mathbf{R}^n) \neq f(\mathbf{R}^n)$. This, in turn, implies that $\neg[\mathbf{R}^n \mapsto f(\mathbf{R}^n)]$ is consistent with **ADD**, **EMB**, and $\mathcal{B}$, which is a contradiction to the soundness of propositional logic. Thus, any proof of $[\mathbf{R}^n \mapsto f(\mathbf{R}^n)]$ using the axiomatization $\mathcal{S}_{\mathrm{emb}} = \mathcal{S}(f, h, g, \circ, V, S)$ will use at least $d$ axiom instances of type **INIT**. Now, any proof of $[\mathbf{R}^n \mapsto f(\mathbf{R}^n)]$ in an asymptotically weaker axiomatization can be translated into a proof in $\mathcal{S}_{\mathrm{emb}}$ with at most a constant factor blow-up in length. Thus, the proof using the asymptotically weaker axiomatization must have length $\Omega(d)$.

## 4.3 Implications for Prominent Voting Rules

In this section, we apply the general bound of Theorem 2 to several existing axiomatizations of important voting rules. We start with the axiomatization of the Borda rule that we used in Theorem 1.

There, we showed that the outcome of the Borda rule at any profile can be explained in $O(m^2)$ steps. We can now show that this is asymptotically tight.

**Corollary 1.** *With high probability, the outcome of the Borda rule on a random profile $\mathbf{R}^n$ requires $\Omega(m^2)$ steps to explain assuming the $\mathcal{L}$-axiomatization $\mathcal{S}_{\text{Borda}}$.*

*Proof sketch.* Let $\mathcal{S}(f, h, g, \circ, V, S)$ be the $\mathcal{L}$- axiomatization of the Borda rule, where $V = \mathbb{Q}^{\binom{m}{2}}$, $h(\mathbf{R}) = \delta^{\mathbf{R}}$ and $g(\delta) = \arg\max_{a \in \mathcal{A}} \sum_{b \in \mathcal{A}} \delta_{ab}$. Because Borda satisfies reinforcement, $g$ admits intersection. We let $S$ be the set consisting of elementary profiles, of cyclic profiles (as defined in Section 3), and of cancellation profiles: a cancellation profile is one in which for each pair $a, b$, there is an equal number of voters preferring $a$ to $b$, and preferring $b$ to $a$. Note that $S$ is made up of a finite number of elementary and cyclic profiles, plus an infinite number of cancellation profiles which are all mapped to the all-zero delta vector by $h$. Hence, $S$ satisfies the condition of being a finite union of sets whose image under $h$ is contained in a one-dimensional subspace.

We need to show that this axiomatization is asymptotically weaker than the axiomatization described in Section 3. Then Theorem 2 implies the desired result, since the dimension of $V$ is $\Theta(m^2)$.

Each **ELEM**, **CYCL**, and **CANC** axiom is an **INIT** axiom. Each **REINF** axiom is an **ADD** axiom. Each **MULT** axiom can be deduced by repeatedly applying an **ADD** axiom. Each **REINF-SUB** axiom can be inferred by combining several **ADD** axioms using propositional reasoning; similarly **SIMP** axioms can be inferred from **ADD** axioms. The details are in Appendix C.2. □

Next, we consider the *plurality rule*. Under this rule, the winning alternatives are those that are ranked in first position by the largest number of voters. Sekiguchi [2012] has given a characterization of this rule (based on an earlier result of Yeh [2008]), using the axioms anonymity, neutrality, reinforcement, faithfulness, and tops-only. Inspecting the proof, we see that the full neutrality axiom is not needed, and only the *orbit axiom* [Brandt and Geist, 2016] is required; this axiom stipulates that symmetric alternatives must either be all winning or all losing. Faithfulness requires that if there is only one voter, then the only winning alternative is the top choice of that voter. Tops-only requires that if in two different profiles $\mathbf{R}_1$ and $\mathbf{R}_2$ defined on the same voters, each voter ranks the same alternative as top choice in both profiles, then $f(\mathbf{R}_1) = f(\mathbf{R}_2)$. Using appropriate fomalizations of the axioms, is possible to translate Sekiguchi's [2012] proof into an explanation of the plurality outcome in our propositional language, and the resulting proofs will be of length $O(m)$. Using Theorem 2, we can show that this is tight. The proof of the following corollary can be found in Appendix D.

**Corollary 2.** *With high probability, the outcome of plurality rule on a random profile $\mathbf{R}^n$ requires $\Omega(m)$ steps to explain, under the axioms anonymity, reinforcement, orbit, faithfulness, and tops-only.*

Throughout this paper, we have discussed voting rules whose input is specified by profiles of rankings. An alternative paradigm is used by *approval voting* [Brams and Fishburn, 2007], which allows voters to indicate, for each alternative, whether they approve or disapprove it. Then, formally, the input to the voting rule is a profile of subsets (of approved alternatives), rather than rankings. Approval voting declares that those alternatives that have been approved by the highest number of voters are winners. This rule has been axiomatically characterized among voting rules with this input format by Fishburn [1978, 1979]. He gives two axiomatizations, both using axioms similar to previous examples. After redefining the set $\mathcal{R}$ of profiles to use approval ballots, our general lower bound (Theorem 2) can be proven verbatim, and implies an $\Omega(m)$ lower bound for explanations of approval voting obtained using Fishburn's axiomatizations; details are relegated to Appendix E.

**Corollary 3.** *With high probability, the outcome of approval voting on a random profile $\mathbf{R}^n$ of approval ballots requires $\Omega(m)$ steps to explain, under the axioms of anonymity, reinforcement, orbit, faithfulness, disjoint equality, cancellation.*

## 5 Discussion

We wrap up with a discussion of the practical implications of our theoretical results.

First, we wish to emphasize that our main result, Theorem 2, holds with respect to almost every profile. Focusing on Borda as an example, this means that the (computationally-efficient) explanation-generating algorithm given by Theorem 1 not only constructs the (asymptotically) shortest possible explanations in the worst case — it constructs the (asymptotically) shortest possible explanation with respect to almost every profile. That said, since these results are asymptotic, there might be some

benefit in designing a search algorithm that computes the absolutely shortest explanation for any given profile. This appears to be a very difficult computational problem; preliminary experiments suggest that standard heuristic search or mathematical programming techniques cannot be used to directly tackle it.

Second, whether our results should be seen as positive or negative depends on the application. Most group decisions — such as those made through online services like `Robovote.org` — involve only a few alternatives: restaurants, movies, vacation spots, best paper awards (from a shortlist), or even prototypes to develop. In these cases, an explanation of linear or quadratic length is perfectly reasonable, so our results should be viewed in a positive light.

By contrast, some settings involve many alternatives. For example, in the work of Lee et al. [2019] — where the Borda rule is used to aggregate predicted preferences — the set of alternatives consists of hundreds of nonprofit organizations that may receive an incoming food donation. In this case, an explanation of quadratic length is a nonstarter, although linear-length explanations (such as those available for plurality and approval) may be viable. The good news is that Theorem 2 can help identify new axiomatizations that lead to short explanations, by providing necessary conditions; to explain Borda outcomes when there are hundreds of alternatives, we must find a new axiomatization that does not take the form of the axioms in Section 4.1 or that uses a different embedding.[3] Our hope is that these insights will lead to new results in social choice theory, which could, in turn, be used to design explainable AI systems that are currently beyond reach.

## Broader Impact

Our work is motivated by societal applications of voting and virtual democracy, as we discuss in Sections 1 and 5. We expect our work to ultimately make these applications more transparent and trustworthy.

We do not foresee negative consequences for our work. It is particularly noteworthy that our explanation approach is not influenced by biases in data, as it builds purely on uncontroversial axiomatic properties rather than deriving explanations from data.

That said, we acknowledge that the virtual democracy approach itself (which is outside the scope of this paper) faces many ethical challenges, including questions about who gets to participate and whether participants' biases influence preference models, thereby negatively affecting aggregate decisions or recommendations made by the system.

## Acknowledgement

We have benefited from detailed and careful comments of the anonymous reviewers. This work was partially supported by the National Science Foundation under grants CCF-2007080, IIS-2024287 and CCF-1733556; and by the Office of Naval Research under grant N00014-20-1-2488. This work was done in part while Alexandros Psomas was visiting Google Research, Mountain View.

## Footnotes

[1]This definition makes our setting *anonymous*, so that nothing depends on the identity of specific voters. Most of our results apply without this restriction, but we adopt it for ease of exposition.

[2] All results in this section still hold when we replace the operation $\bigoplus$ with any binary operation $\mathcal{R} \times \mathcal{R} \to \mathcal{R}$.

[3]For instance, one could embed the Borda rule into $\mathbb{Q}^m$ via Borda scores, which could lead to an explanation in $O(m)$ steps. However, we are not aware of an axiomatic characterization of Borda that operates in this space and can be used for explanations. One result that comes close is the observation that Borda is the only positional scoring rule that never elects Condorcet losers [Smith, 1973], but that characterization relies on a continuity axiom and implicitly a limit argument that do not fit into our logic framework.

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
