[Supplementary Material]

## A    Explaining the 2009 Burlington Mayoral Election Outcome

The following input profile is derived from election data of the 2009 mayoral election in Burlington, Vermont, which is known to exhibit interesting voting theoretic properties.[4] Five candidates were running. One of them, James Simpson for the Green Party, gathered almost no votes (35 first-place votes compared to 1,306 first-place votes for the next-lowest candidate), so we ignore this candidate for convenience. The other four candidates are

$$K = \text{Bob Kiss} \quad M = \text{Andy Montroll} \quad H = \text{Dan Smith} \quad W = \text{Kurt Wright}.$$

The resulting profile consists of the following 3,352 votes.

| 143 | 37 | 139 | 87 | 48 | 112 | 200 | 55 | 432 | 131 | 50 | 72 | 198 | |
|---|---|---|---|---|---|---|---|---|---|---|---|---|---|
| $H$ | $H$ | $H$ | $H$ | $H$ | $H$ | $K$ | $K$ | $K$ | $K$ | $K$ | $K$ | $M$ | |
| $K$ | $K$ | $M$ | $M$ | $W$ | $W$ | $H$ | $H$ | $M$ | $M$ | $W$ | $W$ | $H$ | $\cdots$ |
| $M$ | $W$ | $K$ | $W$ | $K$ | $M$ | $M$ | $W$ | $H$ | $W$ | $H$ | $M$ | $K$ | |
| $W$ | $M$ | $W$ | $K$ | $M$ | $K$ | $W$ | $M$ | $W$ | $H$ | $M$ | $H$ | $W$ | |

| | 129 | 211 | 89 | 151 | 82 | 114 | 324 | 66 | 89 | 288 | 105 |
|---|---|---|---|---|---|---|---|---|---|---|---|
| | $M$ | $M$ | $M$ | $M$ | $M$ | $W$ | $W$ | $W$ | $W$ | $W$ | $W$ |
| $\cdots$ | $H$ | $K$ | $K$ | $W$ | $W$ | $H$ | $H$ | $K$ | $K$ | $M$ | $M$ |
| | $W$ | $H$ | $W$ | $H$ | $K$ | $K$ | $M$ | $H$ | $M$ | $H$ | $K$ |
| | $K$ | $W$ | $H$ | $K$ | $H$ | $M$ | $K$ | $M$ | $H$ | $K$ | $H$ |

### Step 1

In each of the following profiles, we should declare a complete tie.

| 96 | 96 | 96 | 96 |
|---|---|---|---|
| $H$ | $K$ | $M$ | $W$ |
| $W$ | $M$ | $H$ | $K$ |
| $K$ | $H$ | $W$ | $M$ |
| $M$ | $W$ | $K$ | $H$ |

| 56 | 56 | 56 | 56 |
|---|---|---|---|
| $H$ | $K$ | $M$ | $W$ |
| $K$ | $M$ | $W$ | $H$ |
| $M$ | $W$ | $H$ | $K$ |
| $W$ | $H$ | $K$ | $M$ |

| 160 | 160 | 160 | 160 |
|---|---|---|---|
| $H$ | $K$ | $M$ | $W$ |
| $K$ | $W$ | $H$ | $M$ |
| $W$ | $M$ | $K$ | $H$ |
| $M$ | $H$ | $W$ | $K$ |

If we combine all of the above profiles, we should still declare a complete tie.

| 56 | 160 | 96 | 96 | 56 | 160 | 160 | 96 | 56 | 56 | 96 | 160 |
|---|---|---|---|---|---|---|---|---|---|---|---|
| $H$ | $H$ | $H$ | $K$ | $K$ | $K$ | $M$ | $M$ | $M$ | $W$ | $W$ | $W$ |
| $K$ | $K$ | $W$ | $M$ | $M$ | $W$ | $H$ | $H$ | $W$ | $H$ | $K$ | $M$ |
| $M$ | $W$ | $K$ | $H$ | $W$ | $M$ | $K$ | $W$ | $H$ | $K$ | $M$ | $H$ |
| $W$ | $M$ | $M$ | $W$ | $H$ | $H$ | $W$ | $K$ | $K$ | $M$ | $H$ | $K$ |

### Step 2

In the following profile, the selected winners should be $\{M\}$.

| 948 | 948 |
|---|---|
| $M$ | $M$ |
| $K$ | $W$ |
| $H$ | $H$ |
| $W$ | $K$ |

In the following profile, the selected winners should be $\{M, H\}$.

| 160 | 160 |
|---|---|
| H | M |
| M | H |
| W | K |
| K | W |

Combining the above two profiles, we get that in the following, the winners should be $\{M\}$.

| 160 | 160 | 948 | 948 |
|---|---|---|---|
| H | M | M | M |
| M | H | K | W |
| W | K | H | H |
| K | W | W | K |

In the following profile, the selected winners should be $\{M, H, K\}$.

| 260 | 260 |
|---|---|
| K | M |
| H | H |
| M | K |
| W | W |

Combining the above two profiles, we get that in the following, the winners should be $\{M\}$.

| 160 | 260 | 420 | 948 | 948 |
|---|---|---|---|---|
| H | K | M | M | M |
| M | H | H | K | W |
| W | M | K | H | H |
| K | W | W | W | K |

**Step 3**

Combining the profile of Step 1b (resulting in a complete tie) and the last profile in Step 2 (with winners $\{M\}$), we get that in the following profile, the winners should be $\{M\}$.

| 56 | 160 | 160 | 96 | 260 | 96 | 56 | 160 | 580 | 96 | 948 | 1004 | 56 | 96 | 160 |
|---|---|---|---|---|---|---|---|---|---|---|---|---|---|---|
| H | H | H | H | K | K | K | K | M | M | M | M | W | W | W |
| K | K | M | W | H | M | M | W | H | H | K | W | H | K | M |
| M | W | W | K | M | H | W | M | K | W | H | H | K | M | H |
| W | M | K | M | W | W | H | H | W | K | W | K | M | H | K |

Now consider the input profile, and copy each voter 4 times. This gives

| 572 | 148 | 556 | 348 | 192 | 448 | 800 | 220 | 1728 | 524 | 200 | 288 | 792 | |
|---|---|---|---|---|---|---|---|---|---|---|---|---|---|
| H | H | H | H | H | H | K | K | K | K | K | K | M | |
| K | K | M | M | W | W | H | H | M | M | W | W | H | $\cdots$ |
| M | W | K | W | K | M | M | W | H | W | H | M | K | |
| W | M | W | K | M | K | W | M | W | H | M | H | W | |

| | 516 | 844 | 356 | 604 | 328 | 456 | 1296 | 264 | 356 | 1152 | 420 |
|---|---|---|---|---|---|---|---|---|---|---|---|
| | M | M | M | M | M | W | W | W | W | W | W |
| $\cdots$ | H | K | K | W | W | H | H | K | K | M | M |
| | W | H | W | H | K | K | M | H | M | H | K |
| | K | W | H | K | H | M | K | M | H | K | H |

These two profiles have the same weighted majority margins. Hence the winners of the original profile must be $\{M\}$, which is the output of the Borda rule.

# B Proof of Theorem 2

Before we give the proof, we first introduce a generalization of our framework as presented in Section 4.1: we allow an additional family of axioms (called **PRED**) which is parameterized by a family $T$ of linear functions.

**Definition 5.** *Let $f$ be a voting rule that can be embedded into a linear space $V$ by $h$ and $g$, and assume that $g$ admits operation $\circ$, which is commutative. Let $S \subseteq \mathcal{R}$ be a set of* base profiles*, such that $S$ can be written as a finite union of sets of profiles, $S = \bigcup_{k=1}^{N} S_i$, for some $N$, where each $S_i \subseteq \mathcal{R}$ is a possibly infinite set of profiles, and $h(S_i)$ lies in a* one-dimensional *subspace of $V$. Let $T$ be a (possibly infinite) set of linear functions from $V$ to $\mathbb{Q}$; we refer to these functions as* linear predicates*. Then the $\mathcal{L}$-axiomatization $\mathcal{S}(f, h, g, \circ, V, S, T)$ consists of the following four axioms:*

1. **ADD***: For all $\mathbf{R}_1, \mathbf{R}_2 \in \mathcal{R}$ such that $A_1 \circ A_2 \neq \emptyset$,*
   $[\mathbf{R}_1 \mapsto A_1] \wedge [\mathbf{R}_2 \mapsto A_2] \to [\mathbf{R}_1 \bigoplus \mathbf{R}_2 \mapsto A_1 \circ A_2]$.

2. **EMB***: For all $\mathbf{R}_1, \mathbf{R}_2 \in \mathcal{R}$ such that $h(\mathbf{R}_1) = h(\mathbf{R}_2)$,*
   $[\mathbf{R}_1 \mapsto A_1] \to [\mathbf{R}_2 \mapsto A_2]$.

3. **INIT***: For all $\mathbf{R} \in S$, $[\mathbf{R} \mapsto f(\mathbf{R})]$.*

4. **PRED***: For all $\mathbf{R} \in \mathcal{R}$ and all $t_i \in T$ such that $t_i(h(\mathbf{R})) = 0$,*
   $\bigvee_{A \in g(\mathcal{K}(t_i))} [\mathbf{R} \mapsto A]$, *where $\mathcal{K}(t_i)$ is the kernel of $t_i$.*

The **PRED** axiom is an intraprofile axiom which does not have an analogue among the axioms discussed in Section 3. We include it to give our framework more expressive power, especially for encoding neutrality-type axioms (which require that similar alternatives need to be treated identically). **PRED** encodes the fact that if a profile satisfies some condition (given by $t_i$) then its outcome should reflect it. For example, if $f$ is the Borda rule embedded into $V = \mathbb{Q}^m$ by its scoring function, we can let $T = \{t_{ij} : t_{ij}(v) = v_i - v_j\}$, where $v_i$ is the score of alternative $i$. Then $\mathcal{K}(t_{ij})$ is the set of vectors $v$ with $v_i = v_j$, and thus $g(\mathcal{K}(t_{ij}))$ is the set of voting outcomes $A$ which satisfy $i \in A$ if and only if $j \in A$. In other words, **PRED** would require that any two alternatives with the same Borda score are either both winners or both losers.

In Section 4.1, we stated our lower bound in terms of the dimension of the space $V$. In the enriched model, we state the lower bound in terms of $\dim V$ as well as a complexity measure of the set $T$ of linear predicates. The definition of that measure is admittedly unwieldy, but directly related to the length of explanations.

**Definition 6.** *An outcome $A$ is* uniquely determined *by a subset of linear predicates $T' \subseteq T$ if there is a set of rational numbers $C \subseteq \mathbb{Q}$ such that $A = \bigcap_{t_i \in T'} g(t_i^{-1}(c_i))$, where $t_i^{-1}(c_i) = \{v : t_i(v) = c_i\}$ and $c_i \in C$. The* sensitivity *of $g$ with respect to $T$, $\mathrm{sen}(g, T)$, is the minimum size of $T' \subseteq T$ such that there is an outcome $A$ (of $f$) that is uniquely determined by $T'$. If there is no such $T'$, $\mathrm{sen}(g, T) = +\infty$.*

For the embedding of Borda into $\mathbb{Q}^m$ with $T = \{t_{ij} : t_{ij}(v) = v_i - v_j\}$, we can take the winner set $A = \mathcal{A}$. The function $g$ outputs $\mathcal{A}$ only for vectors $v$ with $v_k = v_\ell$ for all $k, \ell$. Now, $t_{ij}^{-1}(0)$ is the set of vectors $v$ with $v_i = v_j$. Thus, the smallest set $T'$ such that $\bigcap_{t_i \in T'} g(t_i^{-1}(0)) = \mathcal{A}$ is $T' = \{t_{12}, t_{13}, \ldots, t_{1m}\}$. It follows that $\mathrm{sen}(g, T) = m - 1$.

**Theorem 3.** *Let $f$ be a voting rule that can be embedded into a linear space $V$ of finite dimension $d$ by $h$ and $g$. Consider an axiomatization $\mathcal{S}$ of $f$ that is asymptotically weaker than some axiomatization $\mathcal{S}(f, h, g, \circ, V, S, T)$ based on operation $\circ$, base profiles $S$, and linear predicates $T$, satisfying the conditions of Definition 3. Then, with high probability, every explanation of the outcome $f(\mathbf{R}^n)$ at the random profile $\mathbf{R}^n$ using $\mathcal{S}$ consists of $\Omega(\min(d, \mathrm{sen}(g, T)))$ steps.*

Define the random vector $\xi_i = h(R_i)$, where $R_i$ is the ranking (or single-voter profile) associated with the $i^{\text{th}}$ voter (which is selected independently and uniformly at random from all $m!$ possible rankings). Fix an arbitrary basis $B \subseteq V$, and let $c_B(v)$ be the coordinates of a vector $v$ under $B$. Define $X_i = c_B(b)^{\mathsf{T}} c_B(\xi_i)$, for some arbitrary non-zero vector $b \in V$. $X_1, \ldots, X_n$ are i.i.d. random variables with mean $\mu$ and variance $\sigma^2$.

**Lemma 1.** *For any vector $b \neq 0$, the random variable $c_B(b)^{\mathsf{T}} c_B(\xi_i)$ has non-zero mean or non-zero variance.*

*Proof.* We prove that if $c_B(b)^\intercal c_B(\xi_i)$ is a random variable with zero mean then it cannot be deterministically zero (and thus has non-zero variance). Towards a contradiction, $\Pr[c_B(b)^\intercal c_B(\xi_i) = 0] = 1$ implies that $c_B(b)^\intercal c_B(h(\mathbf{R})) = 0$ for every $\mathbf{R} \in \mathcal{R}$. But, by the definition of an embedding, $\{h(\mathbf{R}) : \mathbf{R} \in \mathcal{R}\}$ spans $V$. Therefore $c_B(b)$ must be the all zeros vector; a contradiction. $\qquad\square$

**Lemma 2.** *For any non-zero vector $b$ it holds that $c_B(b)^\intercal c_B(h(\mathbf{R}^n)) \neq 0$ with high probability.*

*Proof.* To prove the lemma we need to show that $\sum_{i=1}^n X_i \neq 0$ with high probability. By Lemma 1 either $\mu \neq 0$ or $\sigma \neq 0$. If $\sigma = 0$, then the $X_i$'s are identical non-zero constants (note that this does not imply that the $\xi_i = h(R_i)$ is a constant vector). We trivially get that $\Pr[\sum_{i=1}^n X_i = 0] = 0$, for all $n$. If $\sigma \neq 0$ then the central limit theorem gives us that

$$\limsup_{n\to\infty} \sup_{z\in\mathbb{R}} \left| \Pr\left[ \sqrt{n} \left( \frac{\sum_{i=1}^n X_i}{n} - \mu \right) \leq z \right] - \Phi\left(\frac{z}{\sigma}\right) \right| = 0.$$

Therefore, for any given $\epsilon > 0$, we have

$$\Pr\left[ \left| \sum_{i=1}^n X_i \right| \leq \epsilon \right] = \Phi\left(\frac{\epsilon - \mu n}{\sqrt{n}\sigma}\right) - \Phi\left(\frac{-\epsilon - \mu n}{\sqrt{n}\sigma}\right) + o(1)$$

$$= \frac{1}{\sqrt{2\pi}} \int_{(-\epsilon-\mu n)/(\sqrt{n}\sigma)}^{(\epsilon-\mu n)/(\sqrt{n}\sigma)} e^{-x^2/2} \, \mathrm{d}x + o(1)$$

$$\leq \frac{1}{\sqrt{2\pi}} \int_{(-\epsilon-\mu n)/(\sqrt{n}\sigma)}^{(\epsilon-\mu n)/(\sqrt{n}\sigma)} 1 \, \mathrm{d}x + o(1)$$

$$= \frac{2\epsilon}{\sigma\sqrt{2\pi n}} + o(1), \tag{1}$$

For any $\delta > 0$ we can pick $\epsilon$ small enough, and $n$ large enough, so that both terms on the right hand side of Equation (1) are smaller than $\delta/2$. It then holds that

$$\delta > \Pr\left[ |\sum_{i=1}^n X_i| \leq \epsilon \right] > \Pr\left[ \sum_{i=1}^n X_i = 0 \right]. \qquad\square$$

**Lemma 3.** *With high probability $h(\mathbf{R}^n)$ does not lie in the subspace spanned by any $d-1$ elements in $h(S)$.*

*Proof.* We can assume that $h(S)$ spans $V$, since otherwise we can add vectors from $V$ to make it so (and the lemma still holds). We start by proving the lemma for finite $h(S)$. We are going to show that, with high probability, the coordinates of $h(\mathbf{R}^n)$ under any basis $B' \subseteq h(S)$ of $V$ have no zero entries. This implies that $h(\mathbf{R}^n)$ is not in any subspace spanned by $d-1$ elements in $h(S)$. To see why this is the case, notice that if $h(\mathbf{R}^n)$ lay in the space spanned by some $B' \subseteq h(S)$, with $|B'| = d-1$, we could add one more $v \in V$ to $B'$ and make it a basis for $V$. Then the coordinates of $h(\mathbf{R}^n)$ with respect to $v$ under this basis would be zero, leading to a contradiction; thus, such a $B'$ cannot exist.

Recall that we have already fixed one a basis for $V$, the basis $B$. For every basis $B' \subseteq h(S)$, there is a unique non-singular matrix $P_{B'}$ such that $B = B' P_{B'}$, and for any $v \in V$, $c_{B'}(v) = P_{B'} c_B(v)$. Thus, it is sufficient to prove that for every basis $B' \subseteq h(S)$ all entries of $P_{B'} c_B(h(\mathbf{R}^n))$ are non-zero with high probability.

For any $B'$, by the union bound, the probability that $P_{B'} c_B(h(\mathbf{R}^n))$ has a zero entry is at most $\sum_{i=1}^d \Pr[P_{B'}^i c_B(h(\mathbf{R}^n)) = 0]$, where $P_{B'}^i$ is the $i^{\text{th}}$ row of $P_{B'}$. Due to its non-singularity, each row of $P_{B'}$ must be non-zero. By Lemma 2 it holds that $\Pr[P_{B'}^i c_B(h(\mathbf{R}^n)) = 0]$ converges to zero as $n$ goes to infinity. By applying the union bound again, we conclude that with high probability $P_{B'} c_B(h(\mathbf{R}^n))$ has no zero entries for every basis $B' \subseteq h(S)$. This concludes the proof for finite $h(S)$.

When $h(S)$ is infinite, we use that $h(S)$ is a union of finitely many one-dimensional subsets, $h(S) = \bigcup_{k=1}^N h(S_i)$. Pick an arbitrary non-zero vector $b_i$ from each $h(S_i)$ and let $\mathbf{B}^* = \bigcup_{k=1}^N \{b_i\}$. Notice that $\mathbf{B}^*$ spans $V$, since each $h(S_i)$ is one dimensional. Therefore, we can use the finite version of this lemma for $\mathbf{B}^*$ and get that $h(\mathbf{R}^n)$ does not lie in the subspace spanned by any $d-1$ elements in $\mathbf{B}^*$.

We claim that $h(\mathbf{R}^n)$ does not lie in the subspace spanned by any $d-1$ elements in $h(S)$ either. Towards a contradiction, assume that $h(\mathbf{R}^n)$ lies in the subspace spanned by $B' \subseteq h(S)$, with

$|B'| = d - 1$. Without loss of generality we assume that the vectors in $B'$ are linearly independent. Then, every element of $B'$ must come from a different $h(S_i)$ (since each $h(S_i)$ is one-dimensional). Let $B' = \{b'_{i_1}, b'_{i_2}, \ldots, b'_{i_{d-1}}\}$ and $h(\mathbf{R}^n) = \sum_{j=1}^{d-1} c_j b'_{i_j}$ where $b'_{i_j}$ is an element of $h(S_{i_j})$, and the $c_j$ are rational numbers. Since each $h(S_i)$ is one dimensional, $b'_{i_j} = q_{i_j} b_{i_j}$ where $q_{i_j}$ is rational and $b_{i_j}$ is the vector we included in $\mathbf{B}^*$. We immediately have that $h(\mathbf{R}^n) = \sum_{j=1}^{d-1} c_j q_{i_j} b_{i_j}$, which implies that $h(\mathbf{R}^n)$ lies in a subspace spanned by $d - 1$ elements in $\mathbf{B}^*$ — a contradiction. $\square$

We are now ready to complete the theorem's proof.

*Proof of Theorem 2.* Let $\mathcal{B} \subseteq S$, $|\mathcal{B}| = d - 1$, be a set of **INIT** axiom instance and let $\mathcal{C}$, $|\mathcal{C}| = \text{sen}(g, T) - 1$, be a set of **PRED** axiom instances. We show that there exists no proof of $[\mathbf{R}^n \mapsto f(\mathbf{R}^n)]$ using the axiomatization $\mathcal{S}_{\text{emb}} = \mathcal{S}(f, h, g, \circ, V, S, T)$ that uses only **INIT** axiom instances in $\mathcal{B}$ and only **PRED** axiom instance in $\mathcal{C}$.

Slightly abusing notation, let $h(\mathcal{B}) = \{v_1, v_2, \ldots, v_{d-1}\}$, and assume without loss of generality that these $d - 1$ vectors are linearly independent. Also assume that $h(\mathbf{R}^n)$ is not in any subspace spanned by $d - 1$ elements of $h(S)$, which happens with high probability by Lemma 3. Therefore, $\{v_1, v_2, \ldots, v_{d-1}, h(\mathbf{R}^n)\}$ forms a linear basis of $V$.[5] Furthermore, since there are at most $\text{sen}(g, T) - 1$ linear predicates in $\mathcal{C}$, by the definition of sensitivity, no winning set $A$ of $f$ can be uniquely determined by the linear predicates in the axioms of $\mathcal{C}$. Thus, we can find a vector $b$ such that $t_i(b) = t_i(h(\mathbf{R}^n))$ for all $t_i \in \mathcal{C}$ and $g(b) \neq g(h(\mathbf{R}^n))$.

For a profile $\mathbf{R}$ we can write $h(\mathbf{R}) = k_1 v_1 + k_2 v_2 + \cdots + k_{d-1} v_{d-1} + k_d h(\mathbf{R}^n)$, for rational $k_1, \ldots, k_d$. Define a new embedding $h' : \mathcal{R} \to V$ by $h'(\mathbf{R}) = k_1 v_1 + k_2 v_2 + \cdots + k_{d-1} v_{d-1} + k_d b$, where $b$ is as above. Due to the uniqueness of this decomposition, $h'$ is well-defined.

Now consider the voting rule $f'$ that outputs $g(h'(\mathbf{R}))$ on a profile $\mathbf{R}$. First, notice that $f$ and $f'$ disagree on $\mathbf{R}^n$, as $f'(\mathbf{R}^n) = g(h'(\mathbf{R}^n)) = g(b) \neq f(\mathbf{R}^n)$ by the choice of $b$. Second, for each profile $\mathbf{R}$ in $\mathcal{B}$, we have $h(\mathbf{R}) = h'(\mathbf{R})$, and therefore $f'(\mathbf{R}) = f(\mathbf{R})$. Third, for two profiles $\mathbf{R}_1, \mathbf{R}_2$ such that $h(\mathbf{R}_1) = h(\mathbf{R}_2)$ we have $h'(\mathbf{R}_1) = h'(\mathbf{R}_2)$, which implies $f'(\mathbf{R}_1) = f'(\mathbf{R}_2)$. Fourth, if $f'(\mathbf{R}) \circ f'(\mathbf{R}') \neq \emptyset$ then $f'(\mathbf{R} \bigoplus \mathbf{R}') = g(h'(\mathbf{R}) + h'(\mathbf{R}')) = f'(\mathbf{R}) \circ f'(\mathbf{R}')$. Finally, $t_i(h(\mathbf{R})) = 0$ implies $t_i(h'(\mathbf{R})) = 0$.

The above facts imply that the new rule $f'$ satisfies **ADD**, **EMB**, $\mathcal{B}$ and $\mathcal{C}$, but that $f(\mathbf{R}^n) \neq f'(\mathbf{R}^n)$. Thus, by the consistency of $\mathcal{L}$ and the soundness of propositional logic, $\mathbf{R}^n$'s outcome cannot be explained assuming $\mathcal{S}_{\text{emb}}$ without using **INIT** axioms outside $\mathcal{B}$ or **PRED** axioms outside $\mathcal{C}$. Since this holds for any $\mathcal{B}$ and $\mathcal{C}$, every explanation of $f(\mathbf{R}^n)$ assuming $\mathcal{S}_{\text{emb}}$ must contain at least $\min(d, \text{sen}(g, T))$ axiom instances of type **INIT** and **PRED**.

Consider a proof of $[\mathbf{R}^n \mapsto f(\mathbf{R}^n)]$ assuming $\mathcal{S}$ of length $r$, which is formally a sequence $\varphi_1, \ldots, \varphi_r$ of formulae. By assumption, $\mathcal{S}$ is asymptotically weaker than $\mathcal{S}_{\text{emb}}$. Thus, for each $\varphi_i$ in the proof which is an axiom instance of $\mathcal{S}$, we can replace $\varphi_i$ by a proof of $\varphi_i$ assuming $\mathcal{S}_{\text{emb}}$. After these replacements, we have obtained a proof of $[\mathbf{R}^n \mapsto f(\mathbf{R}^n)]$ assuming $\mathcal{S}_{\text{emb}}$; let $s$ be the number of intraprofile axiom instances (**INIT** and **PRED**) in this proof. Because we have obtained this proof by performing at most $r$ replacements, each time introducing at most $c$ intraprofile axiom instances, we have $s \leq c \cdot r$. From above, we know that $s \geq \min(d, \text{sen}(g, T))$. Thus, $r \geq \frac{1}{c} \min(d, \text{sen}(g, T))$. Hence, any explanation of the outcome $f(\mathbf{R}^n)$ using the axiomatization $\mathcal{S}$ requires $\Omega(\min(d, \text{sen}(g, T)))$ steps. $\square$

## C   Details for Borda

### C.1   Proof of Theorem 1

It will be useful to define the *beta score* of an alternative $a$ in profile $\mathbf{R}$ as $\beta_a^{\mathbf{R}} = 2b_a \cdot m - m(m-1)$ where $b_a$ is the Borda score of $a$. The beta score is a monotonically increasing linear function of the Borda score. Therefore, selecting the top alternatives based on beta scores or Borda scores defines

the same voting rule. For any profile $\mathbf{R}$ the *beta vector* $\beta^{\mathbf{R}}$ maps alternatives to their beta score. Note that $\beta^{\mathbf{R}}$ is a linear transformation of $\delta^{\mathbf{R}}$. More precisely, define $\hat{\beta}(\delta^{\mathbf{R}})$ as the following beta vector: $\hat{\beta}(\delta^{\mathbf{R}})_a = \sum_{b \in \mathcal{A} \setminus \{a\}} \delta^{\mathbf{R}}_{ab}$.

We begin by establishing two lemmas that relate the length of a explanation in a profile $\mathbf{R}$ to the length of the explanation in a profile with a similar delta vector. Recall that delta vectors are a sufficient statistic to compute Borda outcomes. Therefore, if two profiles $\mathbf{R}_1$ and $\mathbf{R}_2$ have identical delta vectors then they have the same set of winners under Borda. The following lemma shows that given such profiles $\mathbf{R}_1$ and $\mathbf{R}_2$, and the set of winners of one of the two, we can produce a proof of constant length that the other profile has the same set of winners.

**Lemma 4.** *Let $\mathbf{R}_1$ and $\mathbf{R}_2$ be two profiles with the same delta vector. Given that $[\mathbf{R}_1 \mapsto A]$, then $[\mathbf{R}_2 \mapsto A]$ can be explained by* **CANC***,* **REINF** *and* **REINF-SUB** *in $O(1)$ steps.*

*Proof.* Let $\overline{\mathbf{R}_1}$ be the profile with the same voters as $\mathbf{R}_1$, but reversed preferences. Clearly, for all alternatives $a, b \in \mathcal{A}$, $\delta^{\overline{\mathbf{R}_1}}_{ab} = -\delta^{\mathbf{R}_1}_{ab}$. We have the following explanation:

1. $[\mathbf{R}_1 \mapsto A]$

2. $[\mathbf{R}_2 \oplus \overline{\mathbf{R}_1} \mapsto \mathcal{A}]$ (**CANC**)

3. $(1) \wedge (2) \rightarrow [\mathbf{R}_2 \oplus \overline{\mathbf{R}_1} \oplus \mathbf{R}_1 \mapsto A]$ (**REINF**)

4. $[\mathbf{R}_2 \oplus \overline{\mathbf{R}_1} \oplus \mathbf{R}_1 \mapsto A]$ (propositional reasoning from 1–3)

5. $[\mathbf{R}_1 \oplus \overline{\mathbf{R}_1} \mapsto \mathcal{A}]$ (**CANC**)

6. $(4) \wedge (5) \rightarrow [\mathbf{R}_2 \mapsto A]$ (**REINF-SUB**)

7. $[\mathbf{R}_2 \mapsto A]$ (propositional reasoning from 4–6) $\qquad \square$

The next lemma shows a similar fact about sums of profiles.

**Lemma 5.** *Let $\mathbf{R}$, $\mathbf{R}_E$ and $\mathbf{R}_C$ be profiles such that $k_1 \delta^{\mathbf{R}} = k_2 \delta^{\mathbf{R}_E} \oplus \mathbf{R}_C$ for integers $k_1, k_2$, and assume that $[\mathbf{R}_E \mapsto A]$ and $[\mathbf{R}_C \mapsto \mathcal{A}]$. Then $[\mathbf{R} \mapsto A]$ can be explained in $O(1)$ steps.*

*Proof.* The explanation works as follows.

1. $[\mathbf{R}_E \mapsto A]$

2. $[\mathbf{R}_C \mapsto \mathcal{A}]$

3. $(1) \wedge (2) \rightarrow [\mathbf{R}_E \oplus \mathbf{R}_C \mapsto A]$ (**REINF**)

4. $[\mathbf{R}_E \oplus \mathbf{R}_C \mapsto A] \rightarrow [k_2(\mathbf{R}_E \oplus \mathbf{R}_C) \mapsto A]$ (**MULT**)

5. $[k_2(\mathbf{R}_E \oplus \mathbf{R}_C) \mapsto A]$ (propositional reasoning from 1–4)

6. $[k_1 \mathbf{R} \mapsto A]$ (5 and Lemma 4)

7. $[k_1 \mathbf{R} \mapsto A] \rightarrow [\mathbf{R} \mapsto A]$ (**SIMP**)

8. $[\mathbf{R} \mapsto A]$ (propositional reasoning from 6–7)

where the sixth step contains the constant length explanation of Lemma 4. $\qquad \square$

The remainder of the proof focuses on the following task: given a profile $\mathbf{R}$ with $A$ the set of Borda winners, construct and explain two profiles $\mathbf{R}_E$ and $\mathbf{R}_C$ such that (1) $k_1 \delta^{\mathbf{R}} = k_2 \delta^{R_E} \oplus \mathbf{R}_C$, (2) $[\mathbf{R}_E \mapsto A]$, and (3) $[\mathbf{R}_C \mapsto \mathcal{A}]$. Specifically, $\mathbf{R}_E$ will be a sum of elementary profiles whose winner sets have a non-empty intersection, and $\mathbf{R}_C$ will be a sum of cyclic profiles. Our approach borrows ideas and facts from the analysis of the algorithm *Borda-expl* presented by Cailloux and Endriss [2016].

To construct $\mathbf{R}_E$, label the alternatives as $a_1, a_2, \ldots, a_m$ in order of decreasing beta scores, so $\beta_{a_1}^{\mathbf{R}} \geq \ldots \geq \beta_{a_m}^{\mathbf{R}}$. Let

$$\mathbf{R}_E = \bigoplus_{i=1}^{m-1} \mathbf{R}_i,$$

where

$$\mathbf{R}_i = \begin{cases} \frac{\beta_{a_i}^{\mathbf{R}} - \beta_{a_{i+1}}^{\mathbf{R}}}{2} \mathbf{R}_{\text{elem}}^{\{a_1, \ldots, a_i\}} & \text{if } \beta_{a_i}^{\mathbf{R}} - \beta_{a_{i+1}}^{\mathbf{R}} > 0, \\ \mathbf{R}_{\text{elem}}^{\mathcal{A}} & \text{if } \beta_{a_i}^{\mathbf{R}} - \beta_{a_{i+1}}^{\mathbf{R}} = 0. \end{cases}$$

Note that beta scores are always even, so $(\beta_{a_i}^{\mathbf{R}} - \beta_{a_{i+1}}^{\mathbf{R}})/2$ is a non-negative integer.

**Lemma 6.** $[\mathbf{R}_E \mapsto A]$ *can be explained in $O(m)$ steps.*

*Proof.* If $\beta_{a_i}^{\mathbf{R}} = \beta_{a_{i+1}}^{\mathbf{R}}$ for all $i = 1, \ldots, m-1$, then $\mathbf{R}_E$ is composed of copies of the profile $\mathbf{R}_{\text{elem}}^{\mathcal{A}}$. Hence by **MULT** and **ELEM**, we obtain $[\mathbf{R}_E \mapsto \mathcal{A}]$, as required, since in this case $A = \mathcal{A}$.

Otherwise, let $k$ be the smallest index with $\beta_{a_k}^{\mathbf{R}} - \beta_{a_{k+1}}^{\mathbf{R}} > 0$. For each $i = 1, \ldots, m$, an **ELEM** axiom gives

$$\left[ \mathbf{R}_{\text{elem}}^{\{a_1, \ldots, a_i\}} \mapsto \{a_1, \ldots, a_i\} \right].$$

If $\beta_{a_i}^{\mathbf{R}} - \beta_{a_{i+1}}^{\mathbf{R}} > 0$, by **MULT** we have

$$\left[ \frac{\beta_{a_i}^{\mathbf{R}} - \beta_{a_{i+1}}^{\mathbf{R}}}{2} \mathbf{R}_{\text{elem}}^{\{a_1, \ldots, a_i\}} \mapsto \{a_1, \ldots, a_i\} \right].$$

Note that $\{a_1, \ldots, a_i\} \subseteq \{a_1, \ldots, a_{i+1}\} \subseteq \mathcal{A}$. Therefore we can inductively apply **REINF** to combine the first $i$ terms and the $(i+1)^{\text{th}}$ term in the $\bigoplus$-summation in the definition of $\mathbf{R}_E$.

Thus, the outcome of $\mathbf{R}_E$ is

$$\bigcap_{i : \beta_{a_i}^{\mathbf{R}} - \beta_{a_{i+1}}^{\mathbf{R}} > 0} \{a_1, \ldots, a_i\} = \{a_1, \ldots, a_k\}.$$

By choice of $k$, the selected outcome $\{a_1, \ldots, a_k\}$ is the set of alternatives with the highest beta scores, i.e. the set of Borda winners $A$. □

A useful fact, following from the discussion of Young [1974], is that $\mathbf{R}_E$ and $m\mathbf{R}$ have the same beta scores.

**Lemma 7** (Young 1974)**.** *For all alternatives $a \in \mathcal{A}$, $\beta_a^{\mathbf{R}_E} = \beta_a^{m\mathbf{R}}$.*

It remains to construct $\mathbf{R}_C$, and bound the length of its explanation. Lemma 7 implies that $\left( \delta^{\mathbf{R}_E} - m\delta^{\mathbf{R}} \right) \in \mathcal{K}(\hat{\beta})$, where $\mathcal{K}(\hat{\beta})$ is the kernel space of the linear map $\hat{\beta}$ defined above. Cailloux and Endriss [2016] show that the set of delta vectors of all cyclic profiles spans $\mathcal{K}(\hat{\beta})$.

**Lemma 8** (Cailloux and Endriss 2016)**.** *There exists a set of m-cycles $\mathcal{S}, |\mathcal{S}| = \binom{m-1}{2}$, such that $\rho = \{ \delta^{\mathbf{R}_{cyc}^S} : S \in \mathcal{S} \}$ spans $\mathcal{K}(\hat{\beta})$.*

We now have the machinery in place to prove that the profile $\mathbf{R}_C$ has the desired properties.

**Lemma 9.** *There exists a profile $\mathbf{R}_C$ such that $\delta^{\mathbf{R}_C} = k \left( \delta^{\mathbf{R}_E} - m\delta^{\mathbf{R}} \right)$, for some integer $k$, and $\mathbf{R}_C$ is the sum of cyclic profiles. Furthermore, $[\mathbf{R}_C \mapsto \mathcal{A}]$ can be explained in $O(m^2)$ steps.*

*Proof.* By Lemma 8, there exists a basis $\rho$ for $\mathcal{K}(\hat{\beta})$. Let $\delta^{\mathbf{R}_i}$ be the $i^{\text{th}}$ base vector in $\rho$, with $\mathbf{R}_i$ its corresponding cyclic profile ($i \in [\binom{m-1}{2}]$), where a profile $\mathbf{R}_i$ that corresponds to $\delta^{\mathbf{R}_i}$ is guaranteed to exist by Lemma 8.[6] One can therefore decompose the target delta vector as

$$\delta^{\mathbf{R}_E} - m\delta^{\mathbf{R}} = \sum_{i \in [\binom{m-1}{2}]} c_i \delta^{\mathbf{R}_i},$$

where the coefficients $c_i$ are all rationals (since the delta vectors are integer vectors). If there is a negative $c_i$ in this decomposition, we can substitute $\mathbf{R}_i$ by $\overline{\mathbf{R}_i}$, the profile where every voter's preference is reversed; the delta vector changes sign and therefore $c_i \delta^{\mathbf{R}_i} = -c_i \delta^{\overline{\mathbf{R}_i}}$. Thus, without loss of generality, we can assume that all $c_i$ are non-negative.

Next, because all the coefficients are rational, there must be an integer $k$ such that $k \cdot c_i$ is a non-negative integer for all $i \in [\binom{m-1}{2}]$. Let

$$\mathbf{R}_C = \bigoplus_{i=1}^{\binom{m-1}{2}} k \cdot c_i \mathbf{R}_i.$$

We can see that $\delta^{\mathbf{R}_C} = k \left( \delta^{\mathbf{R}_E} - m \delta^{\mathbf{R}} \right)$ as desired.

Towards bounding the length of the explanation, since the profiles $\mathbf{R}_i$ are all cyclic, we can use **CYCL** and **MULT** to show $[kc_i \mathbf{R}_i \mapsto \mathcal{A}]$, for $i \in [\binom{m-1}{2}]$. We can then apply **REINF** $O(m^2)$ times, in any order, to combine these profiles. We conclude that $\mathbf{R}_C$ can be explained in $O(m^2)$ steps. $\quad\square$

Theorem 1 now follows directly from Lemmas 5, 6 and 9. $\hfill\square$

## C.2   Proof of Corollary 1

We finish our proof that the $\mathcal{L}$-axiomatization in the proof is asymptotically weaker than $\mathcal{S}_{\text{Borda}}$.

For convenience, in the following proofs we use the *deduction theorem*, which can be easily proved for this system: if we have given a proof of $\varphi_2$ using $\varphi_1$ as an assumption, then the deduction theorem states that there exists a proof of $(\varphi_1 \to \varphi_2)$ [see Ben-Ari, 2012, Thm. 3.14].

Let $\mathbf{R}, \mathbf{R}'$ be profiles and let $A \subseteq \mathcal{A}$. Consider the **REINF-SUB** axiom instance $([\mathbf{R} \bigoplus \mathbf{R}' \mapsto A] \wedge [\mathbf{R}' \mapsto \mathcal{A}]) \to [\mathbf{R} \mapsto A]$. We show that this axiom instance can be proven using **ADD** axioms:

1. $[\mathbf{R} \bigoplus \mathbf{R}' \mapsto A]$ (assumption)
2. $[\mathbf{R}' \mapsto \mathcal{A}]$ (assumption)
3. For each $B \subseteq \mathcal{A}$ where $B \neq A$:
   (a) $([\mathbf{R} \mapsto B] \wedge [\mathbf{R}' \mapsto \mathcal{A}]) \to [\mathbf{R} \bigoplus \mathbf{R}' \mapsto B]$ (**ADD**)
   (b) $[\mathbf{R} \mapsto B] \to [\mathbf{R} \bigoplus \mathbf{R}' \mapsto B]$ (propositional reasoning from 2 and (a))
   (c) $\neg[\mathbf{R} \bigoplus \mathbf{R}' \mapsto A] \vee \neg[\mathbf{R} \bigoplus \mathbf{R}' \mapsto B]$ (**FUNC**)
   (d) $\neg[\mathbf{R} \bigoplus \mathbf{R}' \mapsto B]$ (propositional reasoning from 1 and (c))
   (e) $\neg[\mathbf{R} \mapsto B]$ (propositional reasoning from (b) and (d))
4. $\bigvee_{C \in \mathcal{P}_\emptyset(\mathcal{A})} [\mathbf{R} \mapsto C]$ (**FUNC**)
5. $[\mathbf{R} \mapsto A]$ (propositional reasoning from 3(e) and 4)
6. $([\mathbf{R} \bigoplus \mathbf{R}' \mapsto A] \wedge [\mathbf{R}' \mapsto \mathcal{A}]) \to [\mathbf{R} \mapsto A]$ (deduction theorem from 1, 2, 5)

Let $\mathbf{R}$ be a profile, let $k \in \mathbb{Z}_+$, and consider the **SIMP** axiom instance $[k\mathbf{R} \mapsto A] \to [\mathbf{R} \mapsto A]$. We prove that this axiom can be proven using the **MULT** axiom, which is easy to deduce from **ADD**.

1. $[k\mathbf{R} \mapsto A]$ (assumption)
2. For each $B \subseteq \mathcal{A}$ where $B \neq A$:
   (a) $[\mathbf{R} \mapsto B] \to [k\mathbf{R} \mapsto B]$ (**MULT**)
   (b) $\neg[k\mathbf{R} \mapsto A] \vee \neg[k\mathbf{R} \mapsto B]$ (**FUNC**)
   (c) $\neg[k\mathbf{R} \mapsto B]$ (propositional reasoning from 1 and (b))
   (d) $\neg[\mathbf{R} \mapsto B]$ (propositional reasoning from (a) and (c))
3. $\bigvee_{C \in \mathcal{P}_\emptyset(\mathcal{A})} [\mathbf{R} \mapsto C]$ (**FUNC**)
4. $[\mathbf{R} \mapsto A]$ (propositional reasoning from 2(d) and 3)
5. $[k\mathbf{R} \mapsto A] \to [\mathbf{R} \mapsto A]$ (deduction theorem from 1 and 4)

# D    Details for Plurality

## D.1    An Upper Bound for Plurality

There are multiple ways of rendering Sekiguchi's [2012] proof in our formal system, where the details depend on the exact formal axioms used. Here we give an axiomatization that leads to particularly simple explanations.

We define a family of base profiles for our axiomatization, consisting of *lollipop profiles* $\mathbf{R}_{\text{lolli}}^A$, for each non-empty $A \subseteq \mathcal{A}$, which has $|A|$ voters. Write $A = \{x_1, \ldots, x_k\}$ and $\mathcal{A} \setminus A = \{y_1, \ldots, y_{m-k}\}$. The first voter has preferences $x_1 \succ x_2 \succ \cdots \succ x_k \succ y_1 \succ \cdots \succ y_{m-k}$, the second voter has preferences $x_2 \succ x_3 \succ \cdots \succ x_k \succ x_1 \succ y_1 \succ \cdots \succ y_{m-k}$, and so on. For example, the profile $\mathbf{R}_{\text{lolli}}^{\{a,b,c\}}$ for $\mathcal{A} = \{a, b, c, d, e\}$ has three votes: $a \succ b \succ c \succ d \succ e$, $b \succ c \succ a \succ d \succ e$ and $c \succ a \succ b \succ d \succ e$. Intuitively, in the profile $\mathbf{R}_{\text{lolli}}^A$, the alternatives in $A$ are symmetric under the cyclic permutation $(x_1\, x_2\, \ldots\, x_k)$, and are all stronger than the other alternatives. Thus, a symmetric and efficient voting rule should select the alternatives in $A$. Note that in $\mathbf{R}_{\text{lolli}}^A$, the alternatives in $A$ each have plurality score 1, and other alternatives have plurality score 0.

We now define our axioms.

1. **LOLLI**: For a lollipop profile $\mathbf{R}_{\text{lolli}}^A$, the set of winners should be $A$. Formally, for each $k \in \mathbb{Z}_+$ $\left[ k\mathbf{R}_{\text{lolli}}^A \mapsto A \right]$.

2. **TOPS**: If the plurality score vectors of two profiles are same, they should select the same winners. Formally, for any profiles $\mathbf{R}_1, \mathbf{R}_2$ with $\alpha^{\mathbf{R}_1} = \alpha^{\mathbf{R}_2}$, $[\mathbf{R}_1 \mapsto \mathcal{A}] \to [\mathbf{R}_2 \mapsto \mathcal{A}]$.

3. **REINF**: For any two profiles $\mathbf{R}_1$ and $\mathbf{R}_2$, and any two subsets of alternatives $A_1$ and $A_2$ with $A_1 \cap A_2 \neq \emptyset$, it holds that $([\mathbf{R}_1 \mapsto A_1] \wedge [\mathbf{R}_2 \mapsto A_2]) \to [\mathbf{R}_1 \bigoplus \mathbf{R}_2 \mapsto A_1 \cap A_2]$.

Let us refer to the $\mathcal{L}$-axiomatization consisting of Axioms $1 - 3$ listed above as $\mathcal{S}_{\text{Plu}}$.

**Theorem 4.** *For any profile $\mathbf{R}$ with $m$ alternatives, the outcome of the Plurality rule can be explained in $O(m)$ steps assuming the $\mathcal{L}$-axiomatization $\mathcal{S}_{\text{Plu}}$.*

*Proof.* Suppose, without loss of generality, that $\alpha_{b_1} \leq \alpha_{b_2} \leq \ldots \leq \alpha_{b_m}$. Then $\mathbf{R}$ can be decomposed into subprofiles as follows:

$$\mathbf{R} = \bigoplus_{k=1}^m \mathbf{R}_i,$$

where $\mathbf{R}_i$ is a profile in which alternatives $b_i, b_{i+1}, \ldots, b_m$ each have plurality score $\alpha_{b_i} - \alpha_{b_{i-1}}$ and the other alternatives have plurality score 0. Write $A = \{b_i, b_{i+1}, \ldots, b_m\}$. Then $\mathbf{R}_i$ has the same plurality score vector as the profile $(\alpha_{b_i} - \alpha_{b_{i-1}})\mathbf{R}_{\text{lolli}}^A$. If $\alpha_{b_i} - \alpha_{b_{i-1}} > 0$, then by **TOPS** and **LOLLI** we have $[\mathbf{R}_i \mapsto \{b_i, b_{i+1}, \ldots, b_m\}]$. By applying **REINF** repeatedly, we then obtain $[\mathbf{R} \mapsto f_P(\mathbf{R})]$ in $O(m)$ steps.    $\square$

We can obtain other similar axiomatizations and upper bounds by replacing the **LOLLI** axiom by other axioms that imply the **LOLLI** axiom. For instance, we can use **ORB** and

- **EFF**: for every profile $\mathbf{R}$ in which each voter ranks $a$ higher than $b$, $\bigvee_{A \in \mathcal{P}_\emptyset(\mathcal{A} \setminus \{b\})} [\mathbf{R} \mapsto A]$

which says that a Pareto-dominated alternative should not be elected. It is easy to check that each axiom instance of **LOLLI** can be deduced from **ORB** and **EFF** using a proof with $O(m)$ steps. Since the explanations in the proof of Theorem 4 contain $O(m)$ instances of **LOLLI**, we can thus produce an explanation of the plurality rule in $O(m^2)$ steps. Similarly, we can deduce instances of **LOLLI** by using **ORB**, **FAITH**, and **MULT** (the latter as defined in Section 3), following the arguments in Sekiguchi [2012, Lemmas 1 and 2]; this again takes $O(m)$ steps per instance of **LOLLI**, giving an overall explanation length of $O(m^2)$. Applying our framework gives a lower bound of $\Omega(m)$ on the proof length both for the axiomatization based on efficiency, and for the one based on faithfulness. Thus, it is conceivable that a different strategy could give shorter explanations under these axiomatizations.

### D.2 Proof of Corollary 2

We prove the result with an additional axioms called *equal support* (this strengthens the lower bound), which says that in a profile where each alternative has either plurality score 1 or 0, the alternatives with score 1 are elected. First we formally define new axioms: orbit, faithfulness, equal support, and tops-only. The *plurality score* $\alpha_c^{\mathbf{R}}$ of an alternative $c \in \mathcal{A}$ in profile $\mathbf{R}$ is the number of voters in $\mathbf{R}$ who rank $c$ in top position. For a bijection $\sigma : \mathcal{A} \to \mathcal{A}$ and a strict order $\succ \in \mathcal{A}!$, write $\sigma(\succ)$ for the strict order obtained from $\succ$ by relabeling alternatives according to $\sigma$, so that $\sigma(a)\sigma(\succ)\sigma(b)$ if and only if $a \succ b$. Given a profile $\mathbf{R}$, write $\sigma(\mathbf{R})$ for the profile with $\sigma(\mathbf{R})(\sigma(\succ)) = \mathbf{R}(\succ)$ obtained from $\mathbf{R}$ by relabeling alternatives according to $\sigma$. Then we say that a profile $\mathbf{R}$ is *invariant* under $\sigma$ if $\mathbf{R} = \sigma(\mathbf{R})$.

- **ORB**: If a profile $\mathbf{R}$ is invariant under a bijection $\sigma : \mathcal{A} \to \mathcal{A}$, and $\sigma(i) = j$, we have $\bigvee_{A \in \alpha_{i,j}}[\mathbf{R} \mapsto A]$ where $\alpha_{i,j} \subseteq \mathcal{P}_\emptyset(\mathcal{A})$ is the set of outcomes such that $\{i, j\} \subseteq A$ or $\{i, j\} \cap A = \emptyset$.
- **FAITH**: If a profile $\mathbf{R}$ contains only a single voter who ranks alternative $a$ first, we have $[\mathbf{R} \mapsto \{a\}]$.
- **EQUAL**: Let $\mathbf{R}$ be a profile in which $\alpha_a^{\mathbf{R}} \in \{0, 1\}$ for all $a \in \mathcal{A}$. Then the alternatives with score 1 are elected, so we have $\left[\mathbf{R} \mapsto \{a \in \mathcal{A} : \alpha_a^{\mathbf{R}} = 1\}\right]$.
- **TOPS**: If the plurality score vectors of two profiles are same, they should select the same winners. Formally, for any profiles $\mathbf{R}_1, \mathbf{R}_2$ with $\alpha^{\mathbf{R}_1} = \alpha^{\mathbf{R}_2}$, $[\mathbf{R_1} \mapsto \mathcal{A}] \to [\mathbf{R_2} \mapsto \mathcal{A}]$.

Formally speaking, Corollary 2 claims a lower bound for the axiomatization consisting of **REINF**, **ORB**, **FAITH**, **EQUAL**, and **TOPS**. Note that the axiomatization does not contain a formal version of anonymity; that axiom is implicit in our formal setup and the definition of $\mathcal{R}$.

*Proof of Corollary 2.* We embed profiles into the linear space $V = \mathbb{Q}^m$, using $h$ which maps a profile $\mathbf{R}$ to the plurality score vector $\alpha^{\mathbf{R}} = (\alpha_a^{\mathbf{R}})_{a \in \mathcal{A}}$ and $g = \arg\max$. The set $S$ consists of all single-voter profiles and of all profiles with $\alpha_a^{\mathbf{R}} \in \{0, 1\}$ for each $a \in \mathcal{A}$. The set $S$ is finite. Further, we use predicates $T = \{t_{ij} : t_{ij}(v) = v_i - v_j\}$. With these predicates, a **PRED** instance requires that in a profile $\mathbf{R}$ in which alternatives $i$ and $j$ have the same plurality score, either both are winners or both are losers. Now assume that the profile $\mathbf{R}$ is invariant under the permutation $\sigma$ with $\sigma(i) = j$. Then we automatically have $\alpha_i = \alpha_j$. Hence, each instance of **ORB** can be inferred from an instance of **PRED**. To calculate the sensitivity $\mathrm{sen}(g, T)$ consider for instance the size of the smallest $T' \subseteq T$ that uniquely identifies the outcome $\mathcal{A}$. Outcome $\mathcal{A}$ only occurs in profiles in which all alternatives have the same plurality score. Note that $t_{ij}^{-1}(0)$ is the set of vectors $v$ with $v_i = v_j$. A smallest set $T'$ such that $\bigcap_{t_i \in T'} g(t_i^{-1}(0)) = \{\mathcal{A}\}$ is $T' = \{t_{12}, t_{13}, \ldots, t_{1m}\}$. Similarly one can show that at least $m - 1$ predicates are required to uniquely determine any other outcome. It follows that $\mathrm{sen}(g, T) = m - 1$. The axiomatization stated in the corollary is asymptotically weaker than the axiomatization derived from the embedding: **FAITH** and **EQUAL** are implied by **INIT**, **ORB** is implied by **PRED**, and **TOPS** is implied by **EMB**. $\qquad\square$

Corollary 2 applies to the axiomatization $S_{\mathrm{plu}}$ used in Theorem 4, since **EQUAL** prescribes the output at any lollipop profile, so any **LOLLI** axiom instance can be deduced from **EQUAL** and **REINF**. Thus, $S_{\mathrm{plu}}$ is asymptotically weaker than the axiomatization in Corollary 2. Hence, explanations using $S_{\mathrm{plu}}$ require $\Theta(m)$ steps.

## E   Details for Approval Voting

### E.1   Proof of Corollary 3

Let us redefine $\mathcal{R}$ to be the set of functions $\mathbf{R} : \mathcal{P}_\emptyset(\mathcal{A}) \to \mathbb{N}$ of profiles of approval ballots; the function $\mathbf{R}$ specifies how many voters submit a given set of approved candidates. With this alternative definition, we can define notions like voting rules $f : \mathcal{R} \to \mathcal{P}_\emptyset(\mathcal{A})$ and our language $\mathcal{L}$ exactly as before. Also, everything in Sections 4.1 and 4.2, and in particular the main lower bound of Theorem 2, continues to apply with the new $\mathcal{R}$. For the distribution over $\mathcal{R}$ used in the definition of "with high probability" for Theorem 2, we can take any distribution $\mathcal{D}$ over $\mathcal{R}$ as long as $h(\mathrm{supp}(\mathcal{D}))$ spans

$V$, for example impartial culture for approval profiles (which selects each voter's approval set i.i.d. uniformly at random).

Now let us define axioms appropriate for the approval-based setting. Given a profile $\mathbf{R}$, the *approval score* of an alternative $a$ is the number of voters who approve $a$.

1. **REINF**: For any two profiles $\mathbf{R}_1$ and $\mathbf{R}_2$, and any two subsets of alternatives $A_1$ and $A_2$ with $A_1 \cap A_2 \neq \emptyset$, it holds that $([\mathbf{R}_1 \mapsto A_1] \wedge [\mathbf{R}_2 \mapsto A_2]) \rightarrow [\mathbf{R}_1 \bigoplus \mathbf{R}_2 \mapsto A_1 \cap A_2]$. (Note that this is identical to the previous definition for strict orders.)

2. **ORB**: If a profile $\mathbf{R}$ is invariant under a bijection $\sigma : \mathcal{A} \to \mathcal{A}$, and $\sigma(i) = j$, we have $\bigvee_{A \in \alpha_{i,j}} [\mathbf{R} \mapsto A]$ where $\alpha_{i,j} \subseteq \mathcal{P}_\emptyset(\mathcal{A})$ is the set of outcomes such that $\{i,j\} \subseteq A$ or $\{i,j\} \cap A = \emptyset$.

3. **FAITH-AV**: If a profile $\mathbf{R}$ contains only a single voter with approval set $A$, we have $[\mathbf{R} \mapsto A]$.

4. **DE**: If a profile $\mathbf{R}$ contains exactly two voters, one with approval set $A$ and one with approval set $B$ where $A \cap B = \emptyset$, we have $[\mathbf{R} \mapsto A \cup B]$.

5. **CANC-AV**: If in profile $\mathbf{R}$ all alternatives have the same approval score, then $[\mathbf{R} \mapsto \mathcal{A}]$.

The voting rule *Approval Voting* (AV) selects the set of alternatives with maximum approval score. To prove our lower bound, similarly to plurality, we embed AV into $V = \mathbb{Q}^m$ using $h$ which maps a profile $\mathbf{R}$ to the vector of approval scores, and $g = \arg\max$. The set $S$ consists of all single-voter profiles, all two-voter profiles with disjoint approval sets, and all profiles in which all alternatives have the same approval score. Then $S$ satisfies the conditions of Theorem 2, because the first two parts are finite, and the third part maps to a one-dimensional subspace of $V$. For the set of predicates, we again take $T = \{t_{ij} : t_{ij}(v) = v_i - v_j\}$ with sensitivity $m - 1$.

The axiomatization with axioms 1–5 above is asymptotically weaker than the axiomatization from Theorem 2: **REINF** follows from **ADD**, **ORB** follows from **PRED**, and **FAITH-AV**, **DE**, **CANC-AV** all follow from **INIT**.

To obtain an upper bound on the length of explanations for AV, one can follow the proofs by Brandl and Peters [2019].

## Footnotes

[4]For instance, the plurality winner ($W$) is different from the winner under Instant Runoff Voting rule ($K$) which Burlington used, and both are different from the Condorcet winner ($M$). See https://en.wikipedia.org/wiki/2009_Burlington_mayoral_election.

[5]In the case that $h(S)$ spans $V' \subsetneq V$, the vectors in $h(\mathcal{B})$ cannot be linearly independent. But, we can still create a basis for $V$ that includes a maximal subset of linearly independent vectors from $h(\mathcal{B})$, $h(\mathbf{R}^n)$ and other vectors from $V$.

[6]The proof of Cailloux and Endriss [2016] gives an explicit construction of $\rho$, thus we can find the profiles $\mathbf{R}_i$ by solving a linear system.