[Reviews · NeurIPS 2020]

Review 1

Summary and Contributions: This paper considers explanations of voting outcomes as a series of formal proofs based oil a set of axioms. They demonstrate that Borda outcomes can be explained in a small number of outcomes.

Strengths: The paper considers an important problem - that of explainability in an outcome of a non majority based voting system. Having such methods in place could lead to more trust in the outcome of something like a Borda score ranking.

Weaknesses: My main comment is that I am not sure that Neurips is the correct venue of this paper. The results seem interesting, and maybe applicable to interpretable ML - however the methods and results are not really aligned with the Neurips community. It is on the authors to justify why Neurips is an appropriate venue for this work.

Correctness: I struggled to understand the proofs of the results.

Clarity: I found the paper difficult to read - most likely due to a lack of knowledge in this area. I didn’t really understand the proofs of the main results. More examples would have been helpful - for example, Figure 1 needed some more explanation.

Relation to Prior Work: Yes.

Reproducibility: Yes

Additional Feedback: ##### After Author Feedback ###### I have read the author feedback, and the other reviews. I still feel like I have very little confidence in reviewing this fairly, however I am happy to raise my review to a marginal accept. I would urge the authors to communicate these ideas more clearly to the community at large.


Review 2

Summary and Contributions: The paper builds on a recent paper by Cailloux and Endriss'16, which proposes a way to justify outcomes of a particular voting rule by "proofs" of how this particular outcome follows from a specific set of axioms (an particular axiomatization of the voting rule). In this paper the authors consider the *length* of such proofs. They show that the CE'16 axioms can prove any outcome of Borda in O(m^2) steps. The main result is describing an abstract class of axiomatizations (that can apply to a broad class of voting rules), and provide a lower bound for the (maximal and average) proof length using such axiomatizations. ** after rebuttal ** I raised my score a bit, since I understand that there is *something* that makes the INIT axiom you use more reasonable than the INIT' axiom I suggested as a thought experiment. However I am not fully happy with your answer. I think there should be some more formal criterion to decide what are reasonable axioms (e.g. by their description size). In any case I think this requires some more discussion in the paper.

Strengths: The idea is very cool. I also see the potential application to machine learning once the framework becomes more developed: the Borda rule is very simple to explain even without logical proofs (in fact I suspect that most people will find that a "proof" that just adds up candidate scores more convincing). But if we have some black-box function and all we know is that it satisfies some axioms (are there good examples of such functions in machine learning?), then justifying its outcome with a short logical proof could be great.

Weaknesses: , I have serious concerns about the axiomatic system used in the main proof. The length of explanations is tightly related to the specific set of axioms we use. In this respect, the main theorem essentially shows that *for a particular set of axioms* the length is lower bounded by the dimension. But a-priori it is not clear why the set of axioms in Def. 2 is the one we should care about. In particular, the INIT axiom (which is at the heart of the lower bound proof) seems somewhat arbitrary: essentially it is a long table specifying the outcome of every profile in S. Suppose we would add an alternative axiom INIT': \forall R\in \cal R [R -> f(R)] (i.e. that applies to all profiles rather than to profiles in S alone) then clearly we can now provide a proof of length 1 for any profile of any voting rule. Possibly related, it seems like there should be a simple embedding of Borda (and in fact any PSR) to the space N^m that essentially counts the scores of all candidates. It seems that this space would hold properties 1-3 in Def.1. Are there no simple axioms that capture this? (e.g. the INIT axiom could apply to a single voter profile, and then we just need to somehow add all votes. It seems like the length of proofs with such embedding should be \Theta(nm) for all PSRs.

Correctness: seems ok

Clarity: the paper is well written

Relation to Prior Work: I wonder how does Def.1 relates to the definition of generalized scoring rules. Does any of them generalize the other?

Reproducibility: Yes

Additional Feedback: It looks like there should be some way to define/measure the "strength/dimension" of an axiomatic system (E.g. by the description length of the axioms themselves?), and then have "tradeoff theorems" stating that if the axiomatic system is "simple/small/weak" then the proofs must be long (on average/worst case). Theorem 2 shows one point on this fronteer (and the INIT' axiom is another) but it is hard to say why any particular point is important. In that sense I also find Def.3 a bit too specific. Would make more sense to define the "asym. weaker" relation in a way that would apply to any two axiomatic systems and not just to S_emb. Another idea is to measure the strength of the axiom system by the set of voting rules it *can* explain. Intuitively we would expect that if S is sufficiently strong to explain many rules then proofs will be short (as in the case of the INIT' axiom), and thus possibly surprising results like Thm 2 show the existence of systems that are strong enough to explain any rules (all linear-embedding rules?) and yet still requires long proofs. I'm not sure how appealing this direction is though :/ Is there a meaningful sense in which you can treat the axiomatic system and the definition of f (in what language?) as "syntax" and "semantics"? In Def.2 EMB I think you meant [R1->A] -> [R2->A]


Review 3

Summary and Contributions: This paper aims at explaining complicated voting mechanisms using axioms in natural language. The contributions of this paper are all theoretical. The authors first prove that Borda can be explained using at most O(m^2) steps, then give an asymptotic lower bound on the number of steps to explain other voting rules in general.

Strengths: 1. I believe this paper tackles an important problem since it is usually hard to persuade people to use good but complicated voting mechanisms. 2. The theorems in this paper are sound and novel to the best of my knowledge. I also appreciate the proof sketches.

Weaknesses: 1. No empirical examples are provided in this paper. I believe they are unnecessary but simple real-world examples may help improve this paper. 2. Not sure whether the NeurIPS community would appreciate this paper since this paper belongs to an area newly added to NeurIPS.

Correctness: I think so.

Clarity: Yes.

Relation to Prior Work: Yes.

Reproducibility: Yes

Additional Feedback: The paragraph in lines 230-231 is confusing at the first peek. It should be explicit that $\circ$ is an abstract operation. ------------------ after rebuttal ------------------ I have read all reviews and the authors' response. My score remains unchanged.


Review 4

Summary and Contributions: The paper addresses the challenge of explaining why---given the preferences of several voters in an election---a given alternative should win that election. These explanations appeal to first principles about how to treat voters fairly and consistently (so-called axioms) rather than the definition of the voting rule in use (because that voting rule may seem overly complicated or ill-motivated to the person consuming the explanation). The main contribution is a lower bound on the length of such explanations (Theorem 2). The derivation of this result required the development of a novel framework for embedding a voting rule into a vector space relative to a given axiomatisation of that rule. The dimension of that vector space then is the central parameter determining the length of explanations in terms of axioms coming from that axiomatisation. Applications of the general result to three specific and widely-used voting rules are discussed, with most attention being given to the Borda rule. For that rule, the authors also provide a matching upper bound (Theorem 1), showing that we can explain the Borda winner for any given profile of preferences using at most a number of steps that is quadratic in the number of alternatives. This complements an earlier result in the literature, where the existence of such explanations was proved but no bounds on their length had been established.

Strengths: This is a technically strong paper, asking an original question ("can we bound the length of explanations in voting?") addressing an important domain of applications (various forms of online decision support, including democratic decision making). The results are deep and were somewhat surprising to me. I particularly liked the idea of embedding voting rules into vector spaces and the idea of grouping familiar axioms from social choice theory into certain "classes" of axioms (ADD, EMB, INIT, PRED) to be able to prove general results about them. This seems like a very powerful technique that I can imagine will also prove helpful for addressing other problems. The topic of the paper (algorithms for generating explanations for why a given election outcome is the right one), without any doubt, is highly significant. While it certainly is not a typical topic for a paper at NeurIPS, the authors make a convincing case for why it indeed is relevant to the conference. The most direct connection to machine learning is the fact that political elections are not the only scenarios in which we have to aggregate the preferences of several agents, but we face that very same problem also, for instance, in ensemble learning when combining the outputs of several models. Beyond that, the paper connects to the broader discussion of what different subcommunities within the AI community mean when they talk about "explanations". Finally, the paper points to future challenges of an algorithmic nature (computing optimal explanations) that might also be addressed using approaches more commonly discussed at NeurIPS.

Weaknesses: I cannot point to any specific weakness in the paper. Having said this, in my opinion, the division into main paper and supplementary material is not very natural for this kind of topic. My personal preference would have been to reduce the scope of the paper, and to present that part in more depth, ideally avoiding the need for supplementary material altogether. The rest of the material could have been saved for a later journal paper.

Correctness: I have only skimmed through the supplementary material. But I found the proof sketches in the paper itself very useful and believe I managed to get a fairly good impression of how the full proofs might work in detail. Based on this, I have confidence in the technical correctness of the results reported, even if I have not verified them line by line. A few specific remarks regarding very minor points that could be clarified follow. At the very end of Section 2, when you give a concrete example for a sound and complete proof system, it is not immediately obvious to me that the system you give (axioms + tautologies + modus ponens) really is complete. Is it obvious that modus ponens is the only rule required? Maybe a simple reference to a logic textbook will do. In line 287, you write "[this] implies [formula] satisfies ADD...". It's not clear what you mean here (I think a formula cannot "satisfy" an axiom, which is another set of formulas). Should this maybe be "is consistent with"?

Clarity: The paper is very well written and a real pleasure to read. Despite the technically demanding nature of much of the material, the contributions made are explained clearly and precisely. A few remarks regarding minor points that could be clarified further follow. The example presented in Figure 1 is useful, but I still had difficulty going through it before having read the more abstract parts of the paper. For instance, you might want to try and add another half-sentence to better explain in what sense the profiles in Step 1 are "symmetric". Similarly, some additional explanation of what "equivalent restricted to each pair" means might be helpful. Btw, also for someone who knows what this means (such as this reviewer), it is not easy to verify in practice that this kind of equivalence indeed is satisfied. So it might be good to indicate how you suggest a consumer of one of your explanations should go about verifying this for themselves. A suggestion regarding notation: If I'm not mistaken, you think of \mathbb{N} as the set {0,1,2,...}, i.e., it includes the 0. My own preferred use of this symbol is as the set of natural numbers {1,2,3,...}. So I would suggest using \mathbb{N} instead of the slightly awkward \mathbb{Z}_{+}, and to use \mathbb{N} \cup \{0\} in the one (?) place (on page 3) where you actually need {0,1,2,...}. When you explain cyclic profiles, I did not understand what word the letter T is supposed to abbreviate. Why not use A (for "set of *A*lternatives") here, just as for elementary profiles? I suggest to avoid using the symbol \forall to say "for all". This would allow for a clearer distinction between object language (the fragment of propositional logic you are using to describe axioms) and the meta language (the precise use of English to talk about mathematical issues more generally). Indeed, a confused reader might wrongly assume that, in a paper that makes some use of mathematical logic, expressions involving \forall might be formulas in the logical language defined (which of course they are not). In the definitions of axioms on page 5 (and also in a few other places), when you use propositional formulas in a sentence, it is not entirely clear what you mean. A good example is "Formally, [R^A_elem \mapsto A]". What does this mean? Does it mean that this formula is true? Or does it mean that the axiom being defined is the union of all formulas of this kind? This is both a style issue (it's not nice to use propositional formulas as stand-ins for words) and a genuine question about what exactly is being defined here. That is, is an axiom a set of formulas or is it the postulate that a particular set of formulas must evaluate to /true/? I only noticed a single typo in the entire paper: the missing space just before the reference in line 317.

Relation to Prior Work: The discussion of prior work on the problem addressed is adequate. Two remarks on specific issues where an additional reference to related work (addressing entirely different problems) might be warranted follow. Is the MULT axiom the same thing as what is often called the "homogeneity axiom" in social choice theory? If so, you could consider changing the name of the axiom and you should provide a reference. I agree that the definitions in Section 4.1 (embedding of voting rules into vector spaces) appear to be novel. They also are quite interesting. Having said this, reading this section did remind me of work on the compilation complexity of voting rules (the original reference is Chevaleyre, Lang, Maudet, Ravailly-Abadie, "Compiling the Votes of a Subelectorate", IJCAI-2009). The compilation complexity of a voting rule is, loosely speaking, the number of bits required to store an intermediate election result (for part of the electorate) in such a way that the final result can still be computed accurately once the remaining voters have cast their own votes. There seems to be a close connection between the compilation complexity of a rule and the dimension of the vector space required to embed it. Certainly, when trying to prove compilation complexity results, one tends to think of structures very much like your vector spaces. This connections might be worth exploring further, and in any case, you might feel that a reference to this line of work might be in order.

Reproducibility: Yes

Additional Feedback: I'm presenting my remaining comments roughly in order of the paper, rather than ordering them by importance. I do not have any specific questions I would require a response to during rebuttal. The title, "Explainable Voting", in my opinion is overly generic. You might want to consider something more specific, e.g., something that makes a reference to the main contribution (bounds on the length of explanations). When you explain what an axiom instance is, maybe make it explicit that these are (in, I guess, all practically relevant cases) *infinite* sets of formulas. (I assume that this is obvious, but just in case: there are infinitely many profiles, because you do not restrict the number of voters voting in a given profile.) The remark (on page 6) that Borda can also be embedded into a vector space of dimension m (not just m^2) suggests that it might very well be possible to find Borda explanations of linear length---provided a matching axiomatisation of Borda can be found. I think a sentence or two about this would be helpful. This also raises a more general issue. When you informally speak about embeddings, I think you always talk about "embedding a voting rule into a vector space". But if I'm not mistaken, there really are three components to this concept: the embedding of (1) a voting rule into (2) a vector space always takes place wrt to (3) a specific axiomatisation. I think all the formal definitions are perfectly precise about this, but this point could be made clearer in the informal parts of the paper. That is, when you prove a bound on the length of explanations, all you are saying is that for this rule we cannot find shorter explanations using these axioms---but (of course!) if you change the range of axioms you permit yourself to use, you might very well find a shorter explanation. I agree that the use of the impartial culture assumption (page 7) is unproblematic here (even though in other contexts this might raise a red flag). Finally, allow me to comment some more on the significance of the results. I think the general technique for establishing lower bounds we can extract from Theorem 2 is a mathematically deep and potentially important result. Its first application to the Borda rule, yielding Corollary 1, has real practical relevance, as together with Theorem 1 it shows that the method for generating Borda explanations developed in prior work cannot be improved upon (at least not if we stick to the same axioms). This is surprising and interesting. It is great that the general technique developed here can also, pretty much immediately, be applied to other voting rules, and even to approval voting (which takes a different input format from most common voting rules). Having said this, I found Corollary 2 (plurality) and Corollary 3 (approval voting) much less appealing than Corollary 1 (Borda). The reason is that I think some of the axioms used in these results are much less convincing than the Borda axioms. So this is not really a criticism of the paper under review, but rather a criticism of the original work in economic theory where these axioms where first proposed---it just so happens that their shortcomings are particularly evident in the new context of generating explanations. For plurality, the axiom I take issue with is TOPS: This basically says that the winner must be computable in terms of the plurality score. Thus, all the other axioms basically just encode the fact that higher plurality scores are better. For approval voting, the same criticism applies to CANC-AV, which makes direct reference to the approval scores of alternatives. I think an elegant axiomatisation of a voting rule would not have to make direct reference to the way in which scores are computed for that rule. In the specific context of explaining outcomes, I don't find an explanation that involves computing scores for voting rule F to justify the outcome under F very helpful. For the case of approval voting, it might not be so difficult to come up with alternative sets of axioms that are better suited to the task of explanations (and the great thing about the paper is that the general methodology can be applied right away to any new set of axioms). So how about the following axiom: COMBINE: Take any two profiles R and R', with two voters i and j such that R(i) \cap R(j) \not= \emptyset, R'(i) = R(i) \cup R(j), R'(j) = \emptyset, and R'(k) = R(k) for all k \not= i,j. Then, for all sets A, a voting rule should satisfy [R \mapsto A] -> [R' \mapsto A]. In words: if we can get from profile R to profile R' by combining two of the voters in the first profile into a single voter, then both profiles should result in the same outcome. This axiom seems intuitively appealing (so it /looks/ like a weak axiom, which is a good thing). But in mathematical terms it is quite powerful. Indeed, by induction, COMBINE can be used to derive CANC-AV. So it should work with your remaining axioms. And because it is mathematically stronger than CANC-AV, it might even be possible to weaken some of the other axioms (I have not tried). Less crucially, the following axiom also seems appealing to me: BASIC-TIE: If, for some k, there are k voters who each approve of a different single alternative, then the winners should be exactly those k alternatives. I think BASIC-TIE, REINF, and COMBINE would be a suitable axiomatisation for deriving explanations for approval voting winners. I have not checked this carefully, but here is the basic idea: First use BASIC-TIE and REINF to build a "canonical" profile that has the same approval scores as the input profile (by starting with a basic profile in which only the k alternatives with maximal approval score get one vote each, and so forth), and then to use COMBINE (or, of course CANC-AV) to show that this canonical profile must have the same winners as the input profile. ----------------------------------------------------------------------------------------- ADDED AFTER REBUTTAL Thanks for your response to my review. Regarding homogeneity, maybe mention the explanation you gave me also in the paper (I imagine that other readers will have the same question). I wasn't quite sure what you were referring to when mentioning "limit arguments" in your answer to Reviewer 2. This sounds interesting and, space permitting, you may want to expand on this point in the paper.

[Author Response · NeurIPS 2020]

# Author Response: Explainable Voting #8353

**Review 1**

We discuss the impact of our results for machine learning applications in lines 28–48 of the submission. The reflections by Reviewer 4 in points 2 and 8 are a good expression of how we hope to enrich techniques for explainable ML.

**Review 2**

A major question raised by your review is: how much flexibility do we have in choosing our axiom systems? The answer is that we are heavily constrained. It is possible to write down trivial axiom systems such as your INIT$'$ example, but of course these axioms are not normatively appealing. For other axiom systems, it is necessary to establish that they *uniquely characterize* the voting rule in question, since otherwise there will be instances where the algorithm fails to find an explanation. Very few characterization results are known for voting, even for rules other than Borda. (Worse, most known characterizations rely on limit arguments, which does not lead to explanations in our sense.) Arguably, in our paper, we have captured most of the important ones in a single framework.

Given this background, our choice of axioms for the Borda rule is not arbitrary: the axiom system we use (up to minor variations) is the only one known to characterize Borda (without limit arguments), and luckily the axioms used have considerable normative appeal. While, as you note, the INIT axioms could be seen as arbitrary, in the Borda case, they seem well-motivated via simple symmetry arguments.

Regarding using the simple embedding of Borda into space $\mathbb{N}^m$ to get shorter Borda explanations: indeed this would be possible, but in our framework the EMB axiom would then require that $f(R_1) = f(R_2)$ whenever profiles $R_1$ and $R_2$ have the same Borda scores, and this seems too specific. While we don't think this specific embedding will lead to convincing explanations, the thought process you engaged in as a reader is an example of a hope we express in the discussion: "The good news is that Theorem 2 can help identify new axiomatizations that lead to short explanations." In other words, the possibility of new embeddings that would lead to new characterizations and simpler explanations is a clear strength of our general framework, not a weakness.

Regarding the definition of "asymptotically weaker": you are correct that there is a natural version of this definition for general axiom systems. However our definition specific to $\mathcal{S}_{\mathrm{emb}}$ exploits some additional freedom we have in our specific set-up (namely, that we are allowed to use an unlimited number of ADD and EMB axioms), and we need this freedom in our application of the framework to Borda.

Regarding "trade-off theorems" and measuring the "strength" of classes of axiom systems: this is an intriguing idea, but it is not clear how to formalize the notion of a "class" of axiom systems. For the voting context, we are again limited by the small number of axiom systems (with normative appeal) that are known to characterize common voting rules.

**Review 3**

You mention that we do not include empirical examples. For illustration purposes, we did include (in the supplementary material) a sample explanation of Borda applied to the mayoral election in Burlington, VA. Since by their nature the generated explanations follow a common pattern, we did not think it would be instructive to give many more examples. Quantitatively speaking, we did not see sufficiently promising avenues for empirical exploration: since our bounds provably apply to almost all instances, an empirical evaluation will not reveal that shorter explanations are possible in practice. [And this prediction was confirmed by preliminary experiments we have run.]

**Review 4**

We were amazed by your in-depth review, full of great suggestions. Thanks for taking the time to think deeply about our paper. Responding (much too) briefly to some of your points: We will reference a logic textbook as suggested; our (overpowered) proof system generalizes standard Hilbert systems and is thus indeed complete. Yes, "satisfies" should have been "consistent with". Yes, the pairwise equivalence should receive more space in a user-facing explanation; we shortened this too much due to page limit. Agreed, we will avoid using "$\forall$" in the metalanguage. The standard homogeneity axiom is equivalent to the conjunction of MULT and SIMP; we introduced new names since we handle the two parts separately. Thanks for pointing out the resemblence to structures from compilation complexity; we will reference and think more deeply about this. We agree that the tops-only axiom for plurality is unappealing. There are some nicer characterizations of plurality using independence of Pareto-dominated alternatives, but extending our lower-bound technique to the variable-agenda setting will require more work. Your proposed axiomatization of AV is great, and we think it fits into our framework! In particular, along the lines you sketch, it should be possible to derive instances of the cancellation axiom from BASIC-TIE and COMBINE.

[Meta-Review · NeurIPS 2020]

The reviewers agreed that the paper makes a significant contribution to the interesting area of length of proofs / explanations for in social choice. There was a lengthy discussion among the reviewers, largely centering around the normative appeal of axioms and whether the results have the potential to impact explainable ML/AI. The latter point was debated most; let me attempt to summarize. On the one hand, for an end user who wishes an alternative to be chosen according to a fixed set of axioms, then naturally this end user would prefer an explanation solely in terms of those axioms. The paper then bounds the length of these explanations. In particular, showing why a voting mechanism like Borda selected an alternative X would not be sufficient. The explanation should explain why alternative X was chosen, full stop, not why it was chosen by a particular rule; as such, if one invoked the score totals from Borda, one would also need to give the proof that Borda satisfied the given axioms. As the proof that Borda satisfies the axioms is itself quite lengthy, the axiomatic explanation from first principles is much preferred. On the other hand, one may question the likelihood of having such an end user. The normative appeal of some of the given axioms is debatable, especially to the point where an end user would have zero hesitation accepting the steps of the explanation, and the debatable appeal of axioms seems to be pervasive in voting mechanisms. (One may also point out that Arrow's axioms also have normative appeal, which is perhaps why his impossibility result is so famous.) And if the end user did not feel strongly about the axioms, then in lieu of an axiomatic explanation, they might prefer something of the form: "One can show that Borda satisfies the following axioms, which are reasonable, and here are the score totals which led to alternative X being chosen". For Borda, the "proof-by-implementation" explanation is short, though for other voting mechanisms, even this explanation may be longer than a strictly axiomatic one; we could not think of an example, however. A version of the above debate will no doubt take place in the minds of many readers of this paper, especially those from an ML background. As such, we encourage the authors to make a more direct case case for why the explanations studied, and therefore the results given, are practically relevant for the ML community. As a minor note, it took me some time to understand the numbers in Fig 1; perhaps adding "#" to the top row of the tables, or writing "4x" instead of "4", and referencing the extra symbol in the caption, would shorten the time to comprehension for readers.